# neuroWalknet, a controller for hexapod walking allowing for context dependent behavior

**Malte Schilling**[1]*, **Holk Cruse**[2]

1 Malte Schilling, Autonomous Intelligent Systems Group, University of Münster, Münster, Germany,
2 Biological Cybernetics, Faculty of Biology, Bielefeld University, Bielefeld, Germany

* malte.schilling@uni-muenster.de

## Abstract

Decentralized control has been established as a key control principle in insect walking and has been successfully leveraged to account for a wide range of walking behaviors in the proposed neuroWalknet architecture. This controller allows for walking patterns at different velocities in both, forward and backward direction—quite similar to the behavior shown in stick insects—, for negotiation of curves, and for robustly dealing with various disturbances. While these simulations focus on the cooperation of different, decentrally controlled legs, here we consider a set of biological experiments not yet been tested by neuroWalknet, that focus on the function of the individual leg and are context dependent. These intraleg studies deal with four groups of interjoint reflexes. The reflexes are elicited by stimulation of the femoral chordotonal organ (fCO) or groups of campaniform sensilla (CS). Motor output signals are recorded from the alpha-joint, the beta-joint or the gamma-joint of the leg. Furthermore, the influence of these sensory inputs to artificially induced oscillations by application of pilocarpine has been studied. Although these biological data represent results obtained from different local reflexes in different contexts, they fit with and are embedded into the behavior shown by the global structure of neuroWalknet. In particular, a specific and intensively studied behavior, active reaction, has since long been assumed to represent a separate behavioral element, from which it is not clear why it occurs in some situations, but not in others. This question could now be explained as an emergent property of the holistic structure of neuroWalknet which has shown to be able to produce artificially elicited pilocarpine-driven oscillation that can be controlled by sensory input without the need of explicit innate CPG structures. As the simulation data result from a holistic system, further results were obtained that could be used as predictions to be tested in further biological experiments.

## Author summary

Behavior of animals can be studied by detailed observation, but observation alone does not explain the function of the underlying neuronal controller structures. To better understand this function, an important tool can be to develop an artificial structure based on

**Data Availability Statement:** The simulations consist of two main parts: On the one hand, the neuronal controller processing sensory inputs and producing control signals on a per leg basis (the neuroWalknet controller has been implemented in

python (version 3), see https://github.com/hcruse/neuro_walknet and there the 2022 version). On the other hand, a dynamic simulation environment for the body of the hexapod robot Hector [2, 59], which exists as a hardware version and as a dynamic simulation (implementations are publicly available: dynamical simulation environment is realized in C++ and based on the Open Dynamics Engine library, see https://github.com/malteschilling/hector). Here, we use the dynamic simulation.

**Funding:** This research was supported by the research training group "DataNinja" (Trustworthy AI for Seamless Problem Solving: Next Generation Intelligence Joins Robust Data Analysis) funded by the German federal state of North Rhine-Westphalia to MS. The funders had no role in study design, data collection and analysis, decision to publish, or preparation of the manuscript.

**Competing interests:** The authors have declared that no competing interests exist.

simulated neurons and a simulated or physical body. Although typical animal behavior appears complex, the corresponding neuronal structures may be comparatively simple.

The goal for such a hypothetical structure should be to include as many different behaviors as possible, and, at the same time, search for a simple explanation consisting of a minimum of neuronal elements. Furthermore, such a simulation system, e.g., an artificial neuronal network, should contain hypotheses that can be tested in biological experiments.

We propose an extension to such a network, termed neuroWalknet, that is based on a decentralized neuronal structure, using a neural network as a scaffold, that enables various combinations of local neuronal elements that allow for emergent, i.e., not explicitly designed properties. Indeed, neuroWalknet contains further abilities not yet recognized in the earlier version. For instance, neither explicit structures like central pattern generators nor explicit Active Reaction are required to reproduce typical intraleg reactions. Therefore, neuroWalknet presents a holistic approach enabling emergent properties out of the cooperation of small neuronal elements that are context dependent instead of explicit, dedicated elements.

# 1 Introduction

Animals are able to perform a large variety of complex behaviors, which makes it a challenging task to unravel the underlying control structures. One approach is to develop an artificial controller that, for a given set of behaviors, is able to explain the corresponding biological data, while aiming for a simple solution. If successful, the controller might be expanded to include further behavioral elements. Following this line, we earlier proposed a functional control structure, neuroWalknet [1], the focus of which has been on hexapod walking. This controller represents a basic but already complex behavior as it deals with many degrees of freedom.

This network, applied on the robot Hector [2], allows for control of the basic different aspects of forward and backward walking, negotiation of curves, different velocities, walking on irregular ground, climbing, but also those concerning activation of partial or complete deafferentation of legs. In other words, the earlier experiments performed using neuroWalknet focuses on interleg control and we were able to demonstrate that the functional elements of this controller can account for all these behaviors. The different behaviors emerge out of the interaction between the modules of the decentralized control system and with the environment. In this article, it is our goal to further analyze how the proposed structures can explain additional behavioral and neurophysiological findings which would then offer further credence to this hypothetical structure and the underlying principles of that controller.

In particular, there are a number of experimental results focusing on control of the individual leg—eventually addressed as "interjoint reflexes"—, for which no common detailed structure has been developed yet. This stimulates the question to what extent the already existing controller, neuroWalknet, may predict these results, too. To tackle this challenge, in this article we will focus on experiments on intraleg control and will therefore address activation patterns of the three most relevant joints of the legs (Fig 1B), in neuroWalknet called alpha joint (i.e., the Thorax-Coxa joint (T-C joint), the beta joint (i.e. Coxa-Trochanterofemur joint (C-T joint), and the gamma joint (i.e. Femur-Tibia joint (F-T joint)). The corresponding muscle pairs are called Protractor and Retractor, Levator and Depressor, and Flexor and Extensor, respectively. In all addressed cases—all referring to stick insect studies—context dependent reflex changes will be treated that depend on the state of the animal, for example forward vs. backward walking or state swing vs. state stance.

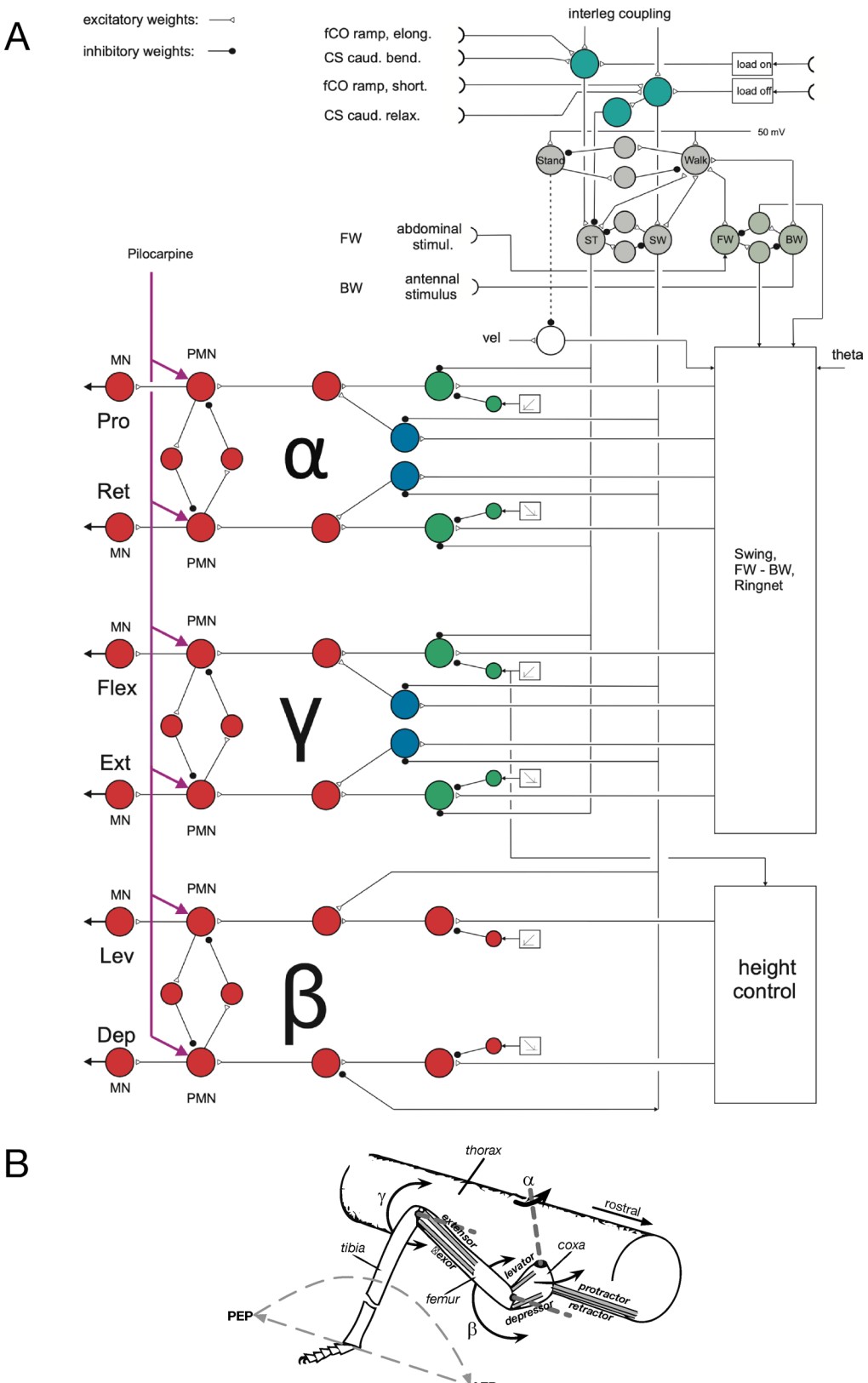

**Fig 1.** A) shows a section of neuroWalknet circuitry including the four units representing the bistable monopole controlling forward-backward walking (FW, BW, stimulated by sensors representing the tactile stimulation at the

abdomen at the back or antennae at the front which induce walking in the opposite direction, i.e., forward walking or backward walking respectively), and another bistable monopole controlling swing-stance (SW, ST) stimulated by load sensor or position sensor (load on, load off). Unit Stance can further be stimulated by two specific sensors, fCO ramp (stimulus elong.), and CS, caudal bending. Correspondingly, unit Swing can be stimulated by fCO (ramp, short.) and CS (caudal, relax.). Unit Walk is recurrently coupled to units Stance, Swing, FW, and BW. For the third bistable monopole Stand-Walk see text. B) shows for a single leg the degrees of freedom as well as the actuating muscles.

Overall, we are considering four groups of experiments (Table 1). The first group of experiments studies reflexes in the alpha joint. Here, we will consider stimulation of campaniform sensilla (CS) by bending the femur to stimulate specific CS. The second and third group of experiments will deal with reflexes triggered by activation of the femoral Chordotonal organ (fCO) relevant for position and velocity of the gamma joint and the response of the gamma joint and beta joint, respectively.

Further, we want to consider a fourth group of experiments in which legs were stimulated by femur bending (CS) or fCO stimulation as in the earlier experiments, but in addition animals were treated with artificially induced pilocarpine, a procedure that elicits oscillations in all three joints of a leg when applied in neurophysiological experiments [3]. A common assumption is that quasi rhythmic leg movements required for walking are basically controlled by local joint oscillators ("central pattern generators", CPG) plus sensory feedback for fine tuning, a view which has been considered as to be supported by the biological experiments addressed in the fourth group. In these experiments, we ask whether the behaviors shown may also be observed if no such dedicated neuronal oscillators were used.

**Table 1. Overview of performed experiments and summary of the biological experiments.**

| Exp. | Stimulus | Biol. Output | Joint | Exp. Var. | State | Observation | Explanation | Res. Fig. | Biol. Ref. |
|---|---|---|---|---|---|---|---|---|---|
| 3.1 | Campaniform Sensilla | Protractor-Retractor | Alpha | Direction | forward | caudal bending –> retractor rostral bending –> protractor | | Fig 2 | Schmitz 1993, Schmitz & Stein 2000, Akay et al. 2004, 2007, Haberkorn et al. 2019) |
| | | | | | backward | caudal bending –> protractor rostral bending –> retractor | reflex reversal dep. on walking dir. | | |
| 3.2.1 | femoral Chordotonal Organ | Flexor-Extensor | Gamma | Direction | forward | flexion of fCO –> flexor | Active Reaction | Fig 3 | Hellekes et al., 2012, Bässler, 1986 |
| | | | | | backward | flexion of fCO –> extensor | reversal | | |
| 3.2.2 | femoral Chordotonal Organ | Flexor-Extensor | Gamma | Curve Walking | inner leg | flexion of fCO –> flexor | Active Reaction | Fig 4 | Hellekes et al., 2012 |
| | | | | | outer leg | flexion of fCO –> const. or ext. | No Active Reaction | | |
| 3.3 | femoral Chordotonal Organ | Levator-Depressor | Beta | Direction | forward | flexion of fCO –> levator | No reflex reversal | Fig 3 | Hess & Büschges, 1999, Bucher et al., 2003 |
| | | | | | backward | flexion of fCO –> levator | | | |
| 3.4.1 | Campaniform Sensilla | Protractor-Retractor | Alpha | application of pilocarpine plus add single sensory stimulus (other connections cut) | | caudal bending –> entrain motor output | | Fig 6 | Akay et al., 2007 |
| 3.4.2 | femoral Chordotonal Organ | Flexor-Extensor | Beta | | | flexion of fCO –> levator | | Fig 7 | Hess & Büschges, 1999 |

In addition, our simulation results contain further predictions that may be tested in new biological experiments. While in the biological experiments usually only motor output of one leg has been recorded, the simulation produces outputs for all three leg joints which closes the loop from simulation study back to neurobiology and provides testable hypotheses for future experiments. The results will support the idea that neuroWalknet [1] indeed represents a powerful basic motor control system which can explain behavioral phenomena based on a simple functional structure.

## 2 Overview of extension of neuroWalknet–A structure for establishing different context

neuroWalknet has been realized as an artificial neural network that acts as a controller for a six-legged robot [1]. Initially, it has been envisioned for control of locomotion for forward and backward walking, but since then has been tested in a variety of behavioral contexts. The main governing principle is the idea of decentralization, which helps simplify the control as the number of explicitly coordinated degrees of freedom is kept low per controller. Such decentralized structures have been widely used for simulation of adaptive walking behavior and have been extended towards planning control architectures [4,5] as well as learning-based approaches [6–10]. In neuroWalknet, decentralization of control is realized through individual controllers for each leg.

In the most recent version (for a detailed explanation see [1] and Fig 1A), each leg controller consists of antagonistic structures as each joint contains antagonistically organized neuronal units with partially linear characteristic and low-pass filter dynamics. This structure allows the separate control of antagonistic "muscles", two for each joint as mentioned above. On the lowest level, units driving the motor output units (MN, Fig 1A, left) are activated by two premotor units (PMN), each connected to another unit (all colored red). Each of these two units receives two branches (green or blue units). Green units represent elements that control the swing movement, the blue units receive different inputs all contributing to stance, which includes input from the so-called ring net that controls the motor output of the alpha joint and the gamma joint during stance. The ring network controls, dependent on the current position of the leg, the activation of the motor units to perform an approximately straight trajectory of the leg tip [1]. Apart from the units marked green or blue, this circuitry is hidden in the box "Swing, FW-BW, Ring" (Fig 1A, lower-right), details of which can be found in Schilling and Cruse ([1], their Fig 2).

In principle, this type of decentralization is shared with the family of earlier Walknet approaches [11–14]. Each leg is controlled by a single controller which decides on which action should be performed by that leg. On a higher level, different actions are represented by so called Motivation Units (Fig 1A, grey units). Motivation units are artificial neurons that represent specific tasks. The current net contains pairs of motivation units as Stand–Walk, Forward–Backward, and Swing—Stance, which are coupled via winner-take-all (WTA) connections (Such structures representing a task might have more than two motivation units, e.g., if several food sources may be represented by different motivation units to decide which of the food sources might be chosen, for example by an ant (e.g., in [15,16]).

A crucial basic property of these motivation unit networks concerns the structure that allows for switching between different contexts (Fig 1A shows a part of a single leg controller, for full structure see [1], their Fig 2), whereby various sensory motor networks (or "sensory-motor primitives") can be combined in different ways and on different levels. The "upper" pair (Fig 1A)) controls the states "Stand" or "Walk", corresponding to passive state or active state (e.g. [17,18]), the two lower pairs control the states "Stance" and "Swing" (Fig 1A, grey units,

ST, SW) and forward walking or backward walking (Fig 1A, grey units, FW, BW). All these pairs represent bistable monopoles, which means that only one unit of each pair can be activated at a given moment in time.

These three pairs of motivation units have already been used in an earlier control network, the so called Walknet, that was however missing the details of the neural structure for control of the individual joints and the antagonistic structure [13,14]. In neuroWalknet [1], to keep the structure simple, instead of using the pair Walk–Stand we used only a single unit ("Leg") as we did not address "Stand" as a behavior in these simulations. In the current extension, we again introduced this pair (Walk–Stand) explicitly as these two paradigms are addressed in some of the biological studies used here (cases concerning state "Stand" are addressed in the Supplement). As illustrated in Fig 1A, activation of unit Stand sets the global velocity to zero (dashed line: inhibitory connection from unit Stand to unit velocity), but additional networks not yet implemented here may be activated as well in the hexapod controller.

As illustrative examples for state Walk, the retractor muscle (Fig 1A, lower left) may be activated by activation of units Forward and Stance, or by activation of units Backward and Swing depending on the activation of the different motivation units.

In the following, we will address specific details of the neuroWalknet structure as relevant for the presented experiments (for further details on neuroWalknet see [1]), i.e., we describe how the experimental situation is reproduced in the simulations.

## Sensory inputs

Input from stimulation of abdomen or antennae—that induce walking in the opposite direction in a stick insect—is, in the simulation, represented by direct activation of motivation units FW or BW (Fig 1A). For sensory input to allow for activating states Swing or Stance, in neuroWalknet [1] a minimal solution had been introduced. Touchdown of the leg tip (Fig 1A, load on) activates unit Stance. Unloading a leg (Fig 1A, load off) activates unit Swing and inhibits Stance (Fig 1A, three turquoise units, input upper right). These units are further influenced by coordination signals that depend on leg position of a neighboring leg. However, in most of the current simulations information from other legs is cut off, i.e., interleg coordination signals are not active as was true for the corresponding biological experiment.

Furthermore, the general, unspecified load signal used in neuroWalknet to record load of the leg during stance phase will not be activated, as in the considered biological experiments the leg is fixed and deafferented apart from the following cases.

Parallel to the "load on" input and "load off" input as applied in neuroWalknet ([1], Fig 2), two additional sensory inputs have been introduced, that have not been used in neuroWalknet. Concerning activation via the CS, a specific input is now introduced that is assumed to be stimulated when the fixed femur is being bent or relaxed ([19–22], Fig 1A, upper left). Bending of femur may not allow a straight forward interpretation of how and why this part of the system contributes to moving the leg during free walking, but the biological experiments show a clear neuronal response when the femur is being bent or relaxed and are therefore well suited for studying the properties of the network. In state forward walking—elicited by abdominal stimulation—caudal bending of the femur activates the retractor, whereas caudal relaxation shows a tendency of activating the protractor. In state backward walking, the relation is inverted. Caudal bending activates the protractor, whereas release of bending provides, however, only weak coupling with the retractor.

Another input to the same units (shown in turquoise in Fig 1A) is assumed to be activated while the femoral chordotonal organ (fCO) receives a ramp-wise elongation or shortening of the fCO (or flexion or extension of the gamma joint) (e.g. [17,23–26]). Such a stimulation can

elicit activation of flexor or extensor of the gamma joint. The reaction depends on velocity of the fCO stimulus and on the state, being passive or active. The latter case could be elicited, for example, by an abdominal tactile stimulation. In passive state, a resistance reflex is observed, whereas in the active state an assistance reflex may be elicited. Again, it is not known in detail how these connections are realized in the biological system, but a ramp stimulus with velocity faster than about 6 mm/s is assumed to lead to a resistance reflex whereas velocities slower than about 2 mm/s usually elicit assistance reflexes [17]. In addition to stimulus velocity, fCO includes position signals from gamma joint as a relevant parameter, as the gamma joint is assumed to influence the beta joint to control body height. Although, resistance reflex and assistance reflex have been interpreted as related structures, separated by a switch from negative to positive connection, here we interpret them as belonging to quite different functional structures. In this study, we focus on the assistance reflex ("active reaction") and will mention the resistance reflex in the Supplement.

### Difference between legs

In the stick insect *Carausius morosus*, the different legs show a somewhat different geometry [27]. In straight forward walking, for the middle leg during the stance phase, initially in the anterior section movement is driven by the retractor muscle (alpha joint) and flexor muscle (gamma joint). In the second part of the stance movement retractor muscle and extensor muscle take over. In backward walking this is inverted: The stance movement starts with protractor and flexor activation followed by protractor and extensor. In principle, this should be as well the case for front and hind legs. However,—due to different anatomy—, in the front leg in forward walking and during stance, only retractor and flexor are used the whole time. In backward walking the protractor and the extensor are used. The hind leg is characterized by a functionally mirrored anatomy compared to a front leg which is also reflected in switched activations. While in principle, the anatomy of the stick insect and of robot Hector are quite similar, there is a difference as in the robot all legs have the same size. In contrast, in the insect front legs and hind legs are longer than the middle leg. Thereby, the structure of the legs of robot Hector are all more similar to a stick insect middle leg. Still, as earlier experiments have focused on front legs in curve walking, our focus will be as well on the front legs in the curve walking experiments [1].

### Fixation of a leg

In the previous neuroWalknet study, the focus has been on coordination of the six legs, whereby in the current study we focus on coordination between joints of a single leg, and with a focus on the stance movement. In contrast to the walking studies presented in the original neuroWalknet study, in all biological experiments considered here, the tested leg is fixed and not coupled to neighboring legs, i.e., neuronal connections to other legs are switched off (apart from five free walking legs used during curve walking [26]) and joints are kept fixed.

This is directly replicated in our simulation experiments. Although the robot is simulated as being in an active state (i.e., walking with 5 legs), in all simulation experiments there is no neuronal connection to the tested leg (i.e., coordination rules are switched off). Further, as in the biological experiments, in the simulation motor units can be recorded, but cannot activate motor output ("muscles") and cannot provide feedback to the sensors, i.e., we deal with open loop experiments in both biological and simulation experiments. Thus, with the leg joints fixed, a complete leg movement cycle cannot be realized. Therefore, in the caudal bending experiments we externally induce a cycle through activation of Stance and Swing.

How can the simulated network be activated? To this end, in the simulation either a FW or a BW unit has to be activated (see Fig 1A), and a stance phase or a swing phase has to be started.

How is the latter realized in the case of caudal bending? In the biological experiments, during normal forward walking, retractor activation starts together with stance. Furthermore, caudal bending in the biological experiments elicits retractor activity. Therefore, in the simulation we coupled caudal bending with state Stance, respectively (see Fig 1A, turquoise units), as we have already done with the less specific load input used in neuroWalknet. To reach a structured result in the simulation experiments and induce a walking cycle, we run caudal bending, i.e., state Stance, for 4 sec, and then enforce a switch to state Swing for another 2 sec (in the figures highlighted as a grey background, for further details see results).

In the biological experiments, there is no comparably strong signal that can be observed during Swing. Rather, during caudal bending activation of retractor appears to decrease in a somewhat irregular way ([20], their Figs 1–4). The swing system of neuroWalknet represents an independent dynamical network to control swing that does not need input (apart from start and ending of the swing controller which we enforce). The 2 sec window has been selected as it provides enough time to finish the Swing procedure and start a new Stance phase. Although we actually only focus on the open loop activation of stance, for illustration, we also use the implementation of a swing phase controller from the original neuroWalknet. In this way, several identical cycles could be run (we will focus in the results on showing example figures for only a single complete cycle).

As for the biological experiments with **caudal bending**, the detailed neuronal structures following the classical fCO stimulation mentioned above are not known. Similarly, we assume that this specific fCO stimulation activates unit Stance (Fig 1A). However, control of the gamma joint is more complicated. Therefore, as addressed above, we focus on the front leg and select four typical cases: walking straight forward or backward, or, during curve walking, using inner leg or outer leg.

For free forward walking, in the stick insect's front leg or middle leg, retractor (alpha joint) and flexor (gamma joint) are activated during at least the first part of the stance period. Therefore, in the simulation of straight forward walking elongation of fCO (flexion of gamma joint) is coupled with unit Stance. This coupling is supported by [24] who found that, during forward walking, elongation of front leg's fCO triggers the retractor. Thus, correspondingly in the neuroWalknet network (Fig 1A), after activation of unit FW and of fCO, unit Stance will activate the retractor (alpha joint) and, due to the structure of the ring net, the flexor (gamma joint) or extensor. Correspondingly, during BW, unit Stance will activate protractor and extensor or flexor. This depends on the sensory position of the alpha joint. If the alpha angle of the leg is pointing to the anterior range, and the controller is in state FW, flexor will be activated, whereas in state BW the extensor will be activated. If the leg is pointing to a more posterior direction, the arrangement is inverted. As above, in the simulation unit Stance will be activated for four seconds and then followed by Swing state for two seconds, because in the biological experiments, similar to the CS data mentioned earlier, the hold sections between the ramp sections may show quite irregular activation. Note, that, due to the open loop situation, during stance only the initial starting values can be shown.

**Curve walking**, is considered as a special case of forward walking, that uses, in principle, the same structure as for straight walking. But in addition, curve walking requires information concerning walking direction, for example, via visual input. Coordination of the different legs, appeared as a rather irregular pattern in biological experiments on curve walking, and is strongly depending on the curve radius [28]. The only biological data showing both temporal structure and trajectories of the legs result from [29], which have then been simulated for free

walking in [1] and which will therefore be used for comparison. Again, as in the biological experiments and the simulations mentioned above, the treated leg is fixed and activated via signals provided by fCO stimulation. Motor output is cut, i.e., we deal with an open loop situation. Stance will be switched off after 4 sec activation and Swing phase will start for 2 sec. In the biological experiments, flexion of fCO leads to flexion of the inner leg, corresponding to an active reaction (flexor is active), whereas such a stimulation of the outer leg shows extensor activation or a weak flexor activation.

In the simulation, the basic information is given by the turning angle (i.e., curve radius) and the leg type, which is then given to the ring net. In our specific case, following the neuronal structure (Fig 1A), fCO stimulation activates unit Stance. During stance, the inner leg is coupled to the flexor while the outer leg is coupled either to the extensor or shows a weak contribution to the flexor depending on the exact value of the curve radius (for leg trajectories of the walking robot see neuroWalknet).

### Variable input representing the effect of Pilocarpine

Another similar, but not identical network is addressed by the four units termed premotor neurons (PMN) in neuroWalknet (Fig 1A, lower left side, and [1], their Fig 2). It represents a soft winner-take-all net which means that the stronger activated unit suppresses the weaker one by lateral inhibition to minimize co-contraction of both antagonistic branches which is, in particular, relevant when the controller switches between branches. This effect is strengthened by using inhibitory units with phasic properties. As a result, if pilocarpine provided by external application is given to both units, the network may show cyclic activation which does, in our simulations, not occur without treatment with pilocarpine. As shown in section Methods, various input values representing the activity of pilocarpine have been tested.

## 3 Results

In the following, we will present results for the four different groups of experiments mentioned in the Introduction (see also Table 1) and show for each case how neuroWalknet deals with the corresponding experimental situations. The results, as presented in Figs 2–4,5, 6 and 7, show simulation results. Sensory inputs are depicted blue. Motor output activation are depicted in red or green color. Continuous lines and dotted lines show motor output that simulates data provided by the biological studies. Dashed lines show motor output that, to our knowledge, has not yet been recorded by experimental studies and may, therefore, be used as predictions for later biological experiments. In Figs 6 and 7, the same colors, now given in black frames, indicate—for the corresponding branches—input via the pilocarpine activation (for details see Methods).

All experiments addressed here—biological as well as our simulation experiments—represent open loop experiments, i.e., the sensory input to CS or fCO is controlled by the experimenter. For closed loop experiments, i.e., free walking as forward or backward or negotiating curves, of robot Hector see [1], their Figs 3, 4, 7 and 8, and corresponding videos.

### 3.1 How stimulation of campaniform sensilla drives retractor and protractor units in the alpha joint depending on context (fw–bw) during active state

**Biological experiments.**   In an interesting paradigm, Schmitz and colleagues [19–22,30] stimulated load sensors—the campaniform sensilla—placed near the beta joint and recorded activation of retractor and protractor of the alpha joint of the middle leg. While the coxa was

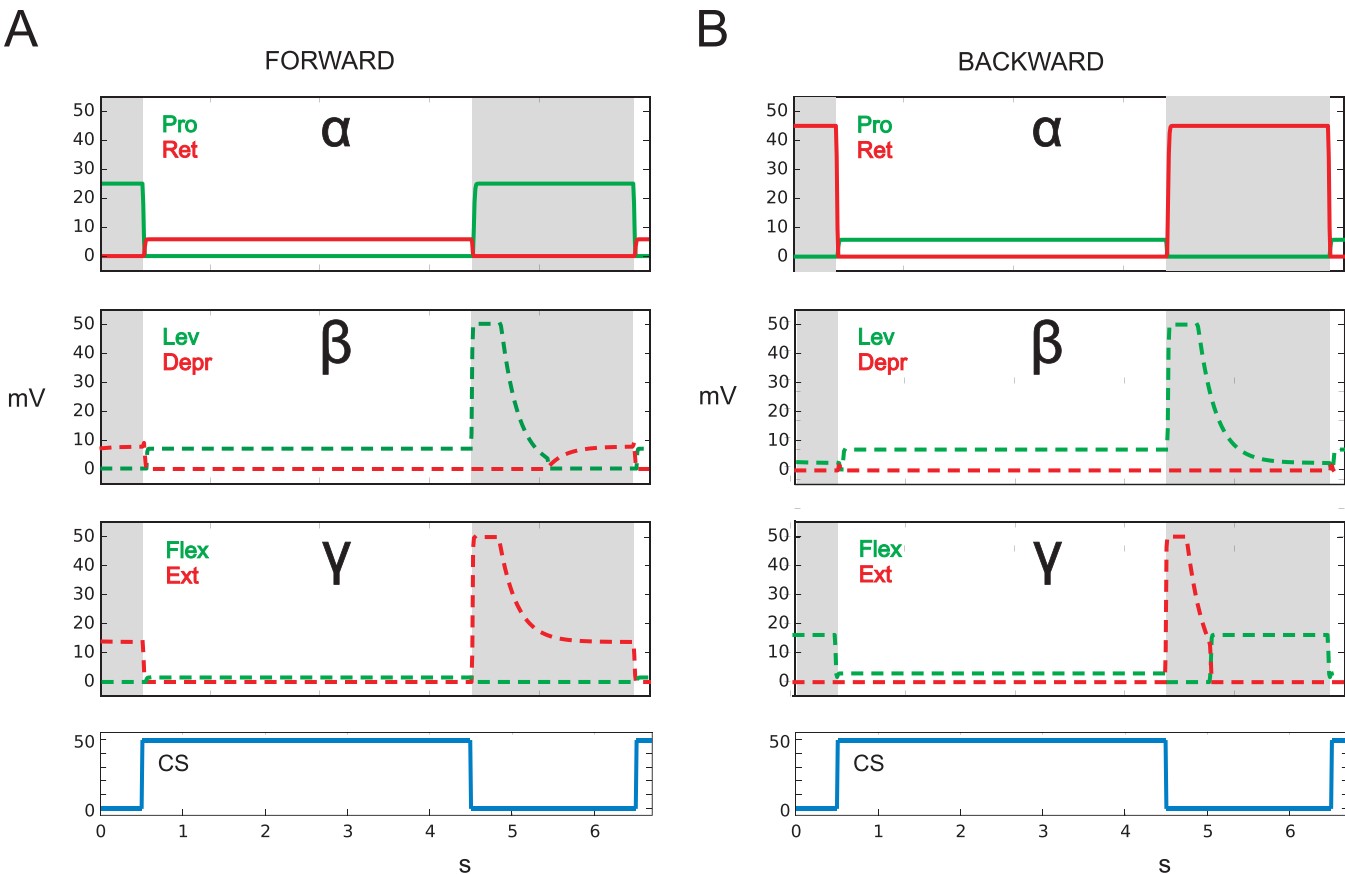

**Fig 2. CS stimulus drives (via caudal bending of femur) protractor or retractor as observed in [20].** Caudal bending, i.e., activation of CS stimulus (below, blue) activates state Stance (highlighted white region), caudal relaxation activates Swing (highlighted grey). A) state: Forward walking, B) state: Backward walking. Abscissa: time (s), ordinate (mV). Dashed lines: simulation data for which no biological results are given. The three output channels are marked by the joints alpha, beta, and gamma.

fixed to the body, and deafferented except for input from the campaniform sensilla, the latter were stimulated by bending the femur caudally or rostrally. When the animal was activated by tactile stimulation to reach an active state, caudal bending elicited activation of retractor muscle, whereas rostral bending elicited protractor activation (in the passive standing animal, the response was in principle the same but showed much weaker activation). [20] extended this paradigm by studying not only middle leg and hind leg, but also front legs. In addition, they introduced context changes by using tactile stimulation of abdomen or antennae to elicit forward or backward walking, respectively, while animals were fixed with legs being removed (or walked on a slippery glass plate). The experiments showed that, with a fixed leg stimulated with femoral bending and the other legs being intact or removed, as above, caudal bending elicits retractor activation during forward walking. But when walking direction is changed to backward walking, caudal bending lead to an immediate switch from retraction to protraction thus showing a context dependent reflex reversal ([20], their Figs 1 and 4). When legs are removed, however, hind legs show an inverse behavior which will not be considered here but dealt with in the Discussion.

**Simulation.** How does neuroWalknet react in these situations? We follow the neuronal structure shown in Fig 1A and assume that the leg tested is in active state. If the femur is bent,

this stimulation triggers the unit Stance (see network in Fig 1A) and thereby inhibits unit Swing. If there is no stimulation by load sensors, the network switches to activate unit Swing.

Therefore, as a result, the simulation with neuroWalknet, when stimulated by caudal bending (Fig 2), indeed shows that when the leg is in state forward, the retractor is activated (Fig 2A, red, filled lines), while when in state backward the same stimulus now activates the protractor (Fig 2B, green, filled lines), as observed in the biological experiments ([20], their Fig 4).

In both simulation experiments, the switch from state Forward to state Backward and back immediately influences the motor output as observed in the biological experiments ([20], their Fig 7).

## 3.2 How stimulation of fCO drives different motor units depending on contexts (negotiation of curves, forward–backward walking) in active state

Biological experiment: Apart from the Resistance Reflex, another intensively studied aspect concerns the so-called **Active Reaction**. In this paradigm [17,23,24,31,32] the animal was fixed and legs had no ground contact. Usually, one leg was restrained and the fCO could be stimulated by a ramp-wise elongation of the fCO apodeme. Following tactile stimulation of the abdomen, which elicits forward walking, elongation of fCO apodeme, i.e., flexion of the gamma joint, elicits flexor activation which is in direct contrast to the resistance reflexes that lead to extensor activation.

If a given elongation has been reached, flexor activation is switched off being replaced by extensor activation. The former part has been called Active Reaction I, the second Active Reaction II, whereby the latter has been interpreted as starting a swing movement. Hellekes et al. [26] continued these studies further. They investigated Active Reaction with most legs being intact and walking on a slippery plane, while only one leg was fixed so that the fCO of this leg could be stimulated experimentally. Forward walking and backward walking as well as curve walking have been studied. First, we will focus on straight walking and compare forward walking and backward walking. In the second part we will consider negotiation of curves as a context which represents quite a complex behavior and where during free walking embodiment plays an important role (for free walking see [1], their Fig 6, and video, and [29,33]).

**3.2.1 Straight walking depending on context (fw–bw).**   Biological experiments: In the following, we consider how the gamma joint may contribute to forward walking and backward walking when these two states are activated by tactile stimulation of abdomen or antennae, respectively. As shown by [26] (their Fig 5), when the fixed leg is in state forward walking, stimulation of the fCO of a front leg (or a middle leg) by elongation elicits an Active Reaction, i.e., activation of the flexor. This behavior would support a stance movement in the case of an intact walking leg. When, however, the animal is stimulated to walk backward, during experimental elongation of fCO apodeme, Active Reaction, i.e., flexor movement, has been observed only rarely. Rather, the extensor is activated instead, which appears functionally sensible because extension would support stance during backward walking (for results concerning the hind leg, see [32], Discussion).

Simulation: How would neuroWalknet react in this situation? When state forward walking is activated and the fCO of a front leg is stimulated by elongation, then—due to the structure of neuroWalknet—the flexor is activated, if the position of the alpha joint points in the anterior position (Fig 3A, green filled lines). Correspondingly, when state backward walking is activated, the extensor is activated (Fig 3B, red filled lines). Both results are in qualitative agreement with results of Hellekes et al. (2012, their Fig 5). The effect is relatively weak due to the smaller angular range used by the robot legs. If in state BW the alpha position points into posterior position, flexor is activated (not shown).

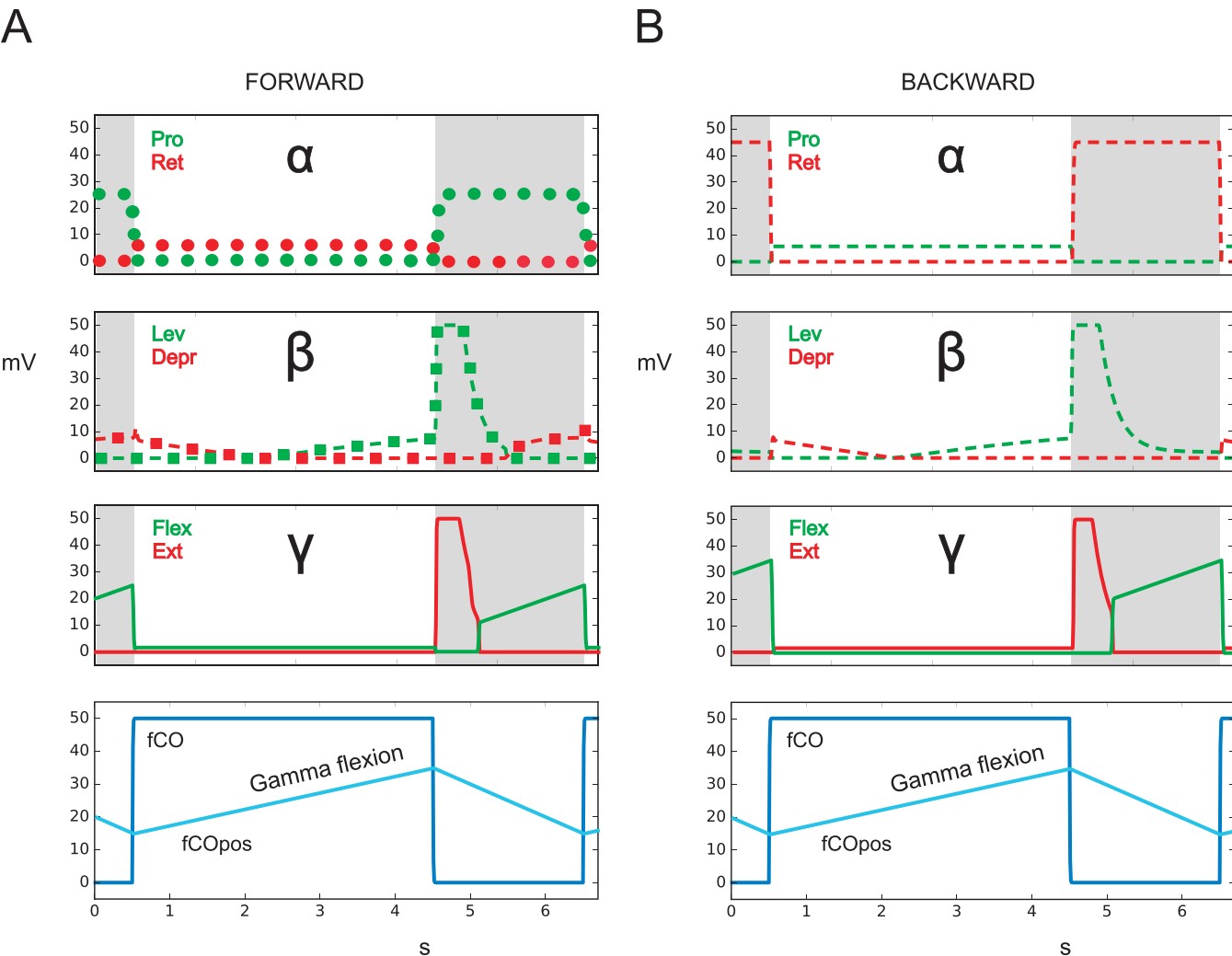

**Fig 3. Active reaction.** fCO stimulus (elongation) drives flexor or extensor as observed in [26], the levator or depressor as observed in (Hess, Büschges 1999, see sect 3.3, red or green dashed squares), and the protractor or retractor as observed by Bässler (1986), red or green dots. Below: activation of fCO stimulus (blue). A) state: forward walking. B) state: backward walking. Abscissa: time (s), ordinate (mV). Dashed lines: simulation data for which no biological results are given. Dotted lines: results from [24]. fCOpos (light blue line) indicates fCO apodeme position, Gamma flexion shows direction of movement. fCO: input to unit swing or unit stance.

Interestingly, [24] performed a detailed study with the front leg of *Cuniculina* and, with fCO stimulation, recorded not only the gamma joint as simulated by [26], but also the alpha joint, i.e., activation of retractor and protractor. He found that in forward walking flexor (gamma joint) activity was accompanied by increase in retractor (alpha joint) activity which, too, agrees with the neuroWalknet simulation results when walking forward (Fig 3A, red dotted lines). Our simulation predicts that changing walking direction to backward walking will activate protractor during stance (Fig 3B, green dashed lines).

**3.2.2 Negotiating curves–context dependent influence on different legs.** Biological Experiments: Besides forward and backward walking, to negotiate curves is a crucial, and probably more difficult, task. In the biological experiments [26], animals have been stimulated by visual input to negotiate curves with five legs walking on slippery ground and with the left middle leg being fixed and prepared to be tested for Active Reaction. When the gamma joint

of the inner leg was experimentally flexed (i.e., fCO apodeme was elongated), an Active Reaction, i.e., a flexor activation and inhibition of extensor was observed. Thus, the observed Active Reaction supports the movement which is expected during stance because flexor action is also observed in the intact inner leg during this case of curve walking. For the outer middle leg the situation is different ([26], their Fig 7). When in the corresponding biological experiments the fCO was elongated, Active Reaction could not be observed ([26], their Fig 7), as the data did neither support a clear flexor activation nor a clear increase of extensor activation. This result might, however, be expected functionally, too, because stance in the outer middle leg does not anymore perform a flexor movement, rather, depending on curve radius, an extension [34].

Simulation: [29] studied curve walking of stick insects using all six legs plus visual stimulation and recorded both footfall patterns as well as leg trajectories. These results show that the trajectories of front leg data are much more pronounced than those of the middle leg. Therefore, these former data have then been used for simulation of negotiation of curves in neuroWalknet (Schilling Cruse 2020, Fig 6). These biological data provided position of the set point (AEP, PEP) of inner and out legs, the local velocities, and the curve radii (turning angle of theta = 75 deg). Therefore, in the current simulation we used data from front legs, where stimulation of fCO was applied to either inner leg or outer leg. As in the former case (Fig 3), Stance, via fCO, was activated for 4 s, followed by a 2 s activation of unit Swing.

In the experiment shown in Fig 4, unit Walk and unit Forward are activated. Fig 4A provides the results for the outer leg. When the fCO is elongated, corresponding to flexion of the gamma joint and unit Stance is activated, a weak extension is observed (red filled lines). For the inner leg, a corresponding simulation is given in Fig 4B. Here elongation of the fCO (i.e., the gamma joint is flexed) leads to an activation of the flexor muscle (green filled lines). Qualitatively, both results agree with the findings of ([26], their Fig 7). Following their interpretation, the inner leg shows active reaction, whereas the outer leg does not. However, these results may also support the idea that the structure of neuroWalknet (combined with, in case of free walking legs, the physical properties of the body) as such explains these data and that specifically dedicated neuronal elements for Active Reaction may not be required. Rather, the behavior called Active Reaction may be considered to emerge out of the complete system.

Taken together, all the experimental results reported by [26] and mentioned here (including data from [24]) are in qualitative agreement with the simulated behavior produced by neuroWalknet. In other words, what has been described as Active Reaction could now be explained as a property emerging from a general, decentralized structure as given in neuroWalknet (see Discussion).

### 3.3 fCO stimulation drives levator units and depressor units, but the active state does not elicit reflex reversal

Biological experiments: In the third group of experiments we turn towards the beta joint. In a series of experiments the influence of stimulating the fCO onto the levator-depressor system (beta joint) has been investigated [35]. In these experiments, front legs and hind legs were fixed. The stimulated middle leg underwent basically the same preparation as above [26]: elongation of fCO, and tactile stimulation of the body to trigger active state. But now activation of levator motor neurons and depressor motor neurons of the beta joint have been recorded. Different to recordings from both systems treated above—the protractor–retractor system or the flexor–extensor system—, in the beta joint no reflex reversal could be observed, i.e., there was no Active Reaction. Rather, elongation of the fCO shows an increasing activation of levator in both states, active as well as passive. There were only some minor differences: Latencies were larger and more variable in the active state. Activation of levator muscles was somewhat

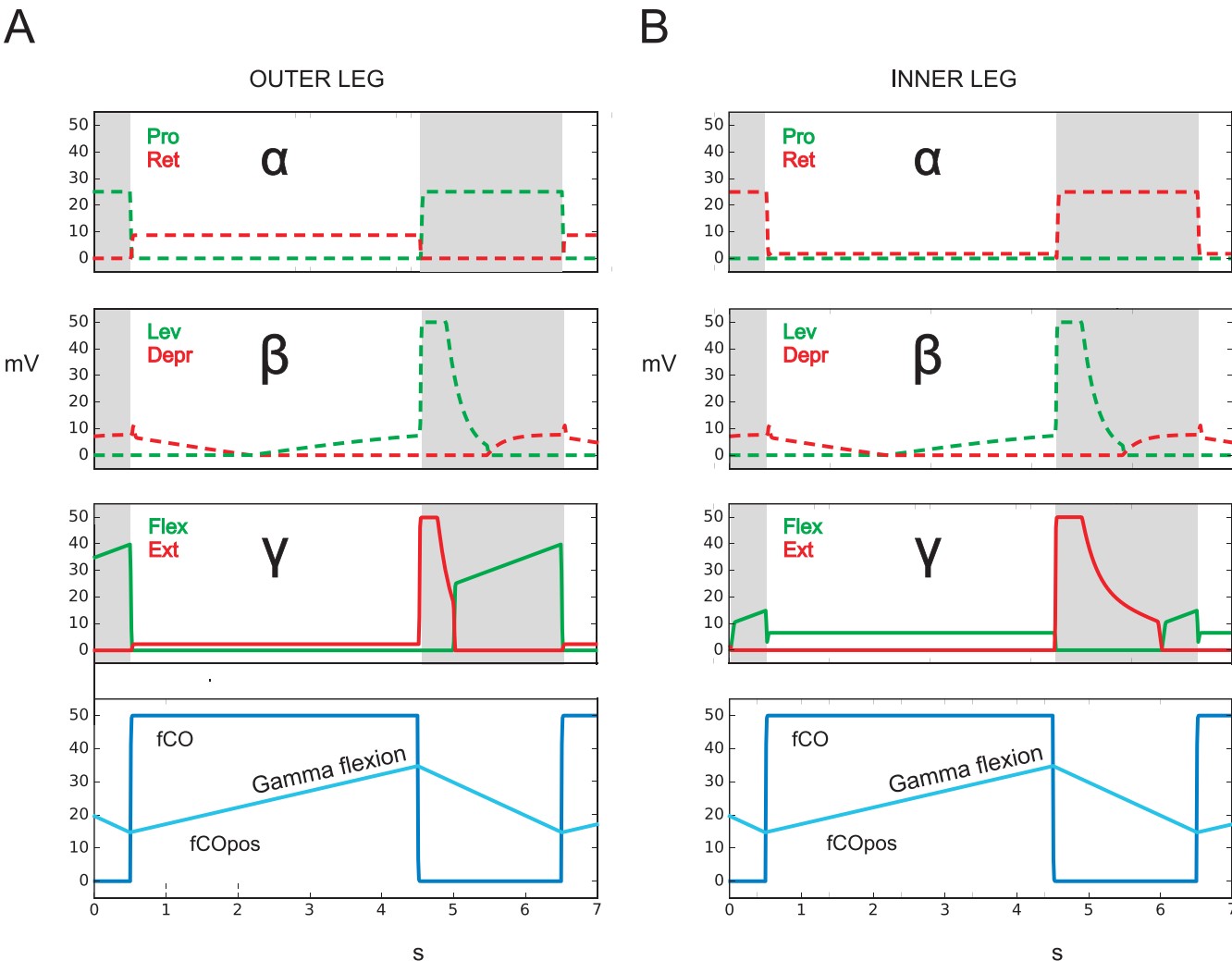

**Fig 4. Negotiation of curves (turn to the right).** fCO stimulus (elongation) drives flexor or extensor as observed in [26], but here front legs are shown. Activation of fCO stimulus (bottom row: blue). A) outer (left) front leg, B) inner (right) front leg. Abscissa: time (s), ordinate (mV). Dashed lines: simulation data for which no biological results are given. fCOpos (light blue line) indicates fCO apodeme position. fCO: input to unit swing or unit stance.

stronger in the active state than in the passive state, but did not show a dependency on several stimulus variables, e.g., position.

In a later study, Bucher et al. [36] provided more quantitative details in the active state. In particular, they found a nonlinear dependency between fCO position and corresponding beta angle values, which was complicated by nonlinear dynamic properties.

Simulation: As in the earlier simulations, we will only focus on active state. In neuroWalknet, the beta controller contributes to height control, the function of which is to keep body-ground distance approximately constant independent of tibia position (as supported by [27]). Therefore, during stance, positional input from gamma joint influences the beta angle (as shown in (Schilling & Cruse, 2020), S1 Fig) and elongation of fCO (flexion) activates the levator as long as—in the case of neuroWalknet—the gamma remains in a range of > 60 deg. As the results show the same type of simulation data as already shown in Fig 3A they are also given in Fig 3A, but the data of the beta joint is highlighted by squared dots.

These results qualitatively agree with data of [35] (their Fig 1A). Note that neuroWalknet is characterized by an antagonistic neuronal structure. Decrease of depressor (red lines) follows an increase of levator (green lines) and both agonists move the output in the same direction.

In these biological experiments, only forward walking has been studied, but following the architecture of neuroWalknet, for backward walking no different influence is expected, as the beta controller in neuroWalknet does not receive input from the forward-backward network ([1], their Fig 2). Thus, Fig 3B (backward walking) shows the same results for the beta joint.

Non-linearities as described by Bucher et al. [36] have not been implemented as these are not required for the small angular range observed during walking [1].

As mentioned in [35] (their Fig 2), latencies are more variable in the active state then in the passive state. This result can be explained, if the neuronal structure follows the principle structure of neuroWalknet. In neuroWalknet, when representing the active state, both the height controller as well as activation via units Swing or Stance can influence the input of levator and depressor. However, in the passive state only part of neuroWalknet, the height controller, may by activated. Therefore, the active state may lead to a higher variability, as observed by [35].

## 3.4 How sensory input (CS, fCO) influence artificially induced rhythmic oscillation

In the fourth group of experiments, we test how oscillations that have been experimentally induced [20,35] can be recreated by neuroWalknet. These and earlier studies have often been seen as an argument for requiring an explicit central pattern generator structure. As such a structure is not fully realized in neuroWalknet, the fourth group of experiment is as well a test if such specific oscillations controlling circuits do exist in the biological system as a necessary element for control of walking or if the present structures in neuroWalknet that do not operate with CPGs, i.e., not treated with pilocarpine, are sufficient to explain such a behavior.

Biological experiment: In the following experiments, legs did not only receive tactile stimulation of abdomen or antennae and CS or fCO stimulation as in the above experiments, but they were in addition treated with pilocarpine. Pilocarpine induces an overall activation that shows internal rhythmic movements of low frequency. In the classical experimental studies (see, e.g., [3]) sensory input and motor output as well as connectives have been cut. In contrast, in more recent experiments as considered here [20,35], sensory stimulation is made possible by leaving the neuronal connections to CS or to fCO intact in order to permit additional sensory rhythmic or arhythmic stimulation.

Importantly, in both of these biological experiments—addressed here and as shown in Fig 6 and Fig 7—the sensory stimulation was rhythmic and characterized by periods much shorter than that of the periods produced by the purely pilocarpine-driven oscillations.

Realization of experimental influence in simulation: How is the experimental effect provided by pilocarpine introduced into neuroWalknet? First of, we consider the neural structure that is present in neuroWalknet. In neuroWalknet, a winner-take-all like structure of competing units can be found in multiple instances. In addition to the competing units, two additional units are used to organize the activations of the other two units through mutual inhibition. These structures, consisting of two excitatory and two inhibitory units, are used in two different contexts. First, one of these structures is present for each joint in each leg network called the premotor neuron net (PMN), controlling the input to the motor neurons (MN) of its joint (units in red, Fig 1A, Fig 5). The second network is employed as part of the motivation unit network, for example, the net that controls the switch between Swing and Stance (SW–ST, grey units in Fig 1A, Fig 5), which is called a bistable monopol. Importantly, there are two differences in the incoming connectivity of these networks. In the motivation network on the leg

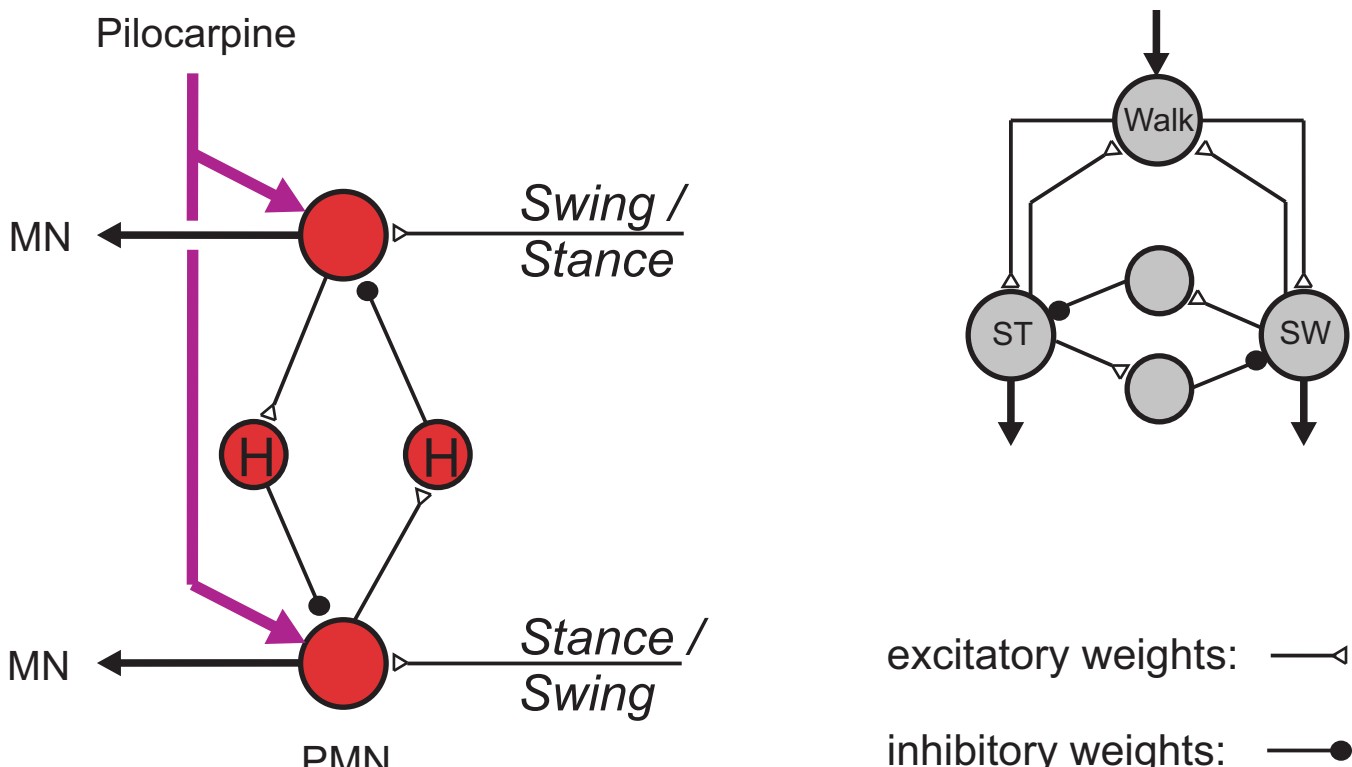

**Fig 5. Two similar, distinct networks that are elements of units shown in Fig 1A.** Left: Four units (red) form a PMN network. Both inhibitory units are equipped with a high-pass filter (H). The purple arrows represent external injection of pilocarpine. Right: four units (grey), two of which are recurrently connected with unit Walk via constant input. Both inhibitory units do not have high-pass properties, but only low-pass filter properties as is the case for all other neurons in neuroWalknet.

level, as in the SW–ST net, both excitatory units receive a stable constant input, which is not the case for the PMN network. Instead, the inhibitory units of the PMN net contain a high-pass filter, i.e., phasic properties, each (Fig 5, marked H), which in turn is not the case for the SW–ST motivation unit net.

While these networks represent two different types, they have in common the property that neither of these networks is able to show oscillations on its own (for more details see [1]). The motivation unit nets are missing the phasic properties, the PMN nets are missing the constant excitatory input as used for the motivation unit structure. This changes for the PMN net, if its excitatory units are activated by additional, constant positive excitation elicited by pilocarpine being provided to the ganglion by external injection. Depending on the time constants of the high-pass filters and the activation strength driving both units, the PMN net may oscillate in different periods and different phase relations. These input channels are shown as purple arrows (Fig 1A, Fig 5) and represent, in the simulation, the strength of the pilocarpine influence to the PMN unit (for details see Methods, Table 2), i.e., an externally and purely experimentally driven additional input. Note, that in the biological experiments this procedure requires surgical treatment.

Details biological experiment 1: In the biological experiment ([20], their Fig 1E, CS period about 4 s, pilocarpine driven period about 3 s), for a forward walking middle leg, caudal bending has driven retractor activation. In this case, indeed the CS signal dominates, i.e., resets the pilocarpine induced oscillation if the CS signal has a somewhat shorter period (i.e., higher frequency) than the pilocarpine driven cycle.

**Table 2. Setting of input parameter and the corresponding periods for the fourth experiment (pilocarpine application).**

| | Protr. | Retr. | Period | Lev. | Depr. | Period | Flex. | Ext. | Period |
|---|---|---|---|---|---|---|---|---|---|
| BSB (s) | 2 | 4–7 | 6–9 | 3.7 | 2.7 | 6 | | | 0–4 |
| Simulation of current Pilocarpine activation | | | | | | | | | |
| input (mV) | 3.0 | 6.0 | | 13.0 | 3.0 | | 5.0 | 5.0 | |
| output Pilo(s) | 3.7 | 6.7 | 10.2 | 2.8 | 1.8 | 4.6 | 2.1 | 2.1 | 4.2 |
| Simulation of Pilocarpine activation used by Schilling and Cruse (2020) | | | | | | | | | |
| input (mV) | 40.0 | 25.0 | | 40.0 | 25.0 | | 50.0 | 50.0 | |
| output Pilo(s) | 6.4 | 5.4 | 11.8 | 2.0 | 5.6 | 7.6 | 2.0 | 2.1 | 4.1 |

Details simulation 1: How can this be simulated? The PMN network, when activated by pilocarpine, elicits independent oscillatory rhythms, one for each joint, that activate the corresponding motor neurons. Different frequencies can be chosen through the variation of input values (Fig 1A, Fig 5, purple arrows, and Table 2). Note, that if these input values are set to zero, no oscillation will occur. As in the biological experiments, connectives to neighboring ganglia have been switched off.

As in the earlier experiments (Sect 3.1), the CS signal (Fig 1A, turquois units) activates unit Stance or unit Swing, which activates further units via the antagonistic branches (Fig 1A, green units and blue units, see also [1]) and thereby in turn influence the PMNs, too. Thus, both inputs—one produced by the externally injected pilocarpine, the other by the sensory signals—are combined in the PMN structure, and the result will then be projected to the motor neurons.

The motor output provided by the network without using pilocarpine is already shown in Fig 2. To better understand as to how these two inputs are integrated in the PMN structure, we show two additional graphics. Fig 6 combines these two graphics where, for each of the three joints, the outputs are plotted pairwise on top of each other. The respective upper plot shows the motor output data for each joint when driven by pilocarpine only, i.e., without sensory input (upper graphics, marked by rectangles filled red or green and framed black, and marked by 'Piloc'). To this end, different periods (selected to be similar to frequencies used by [1]) are used (10.2 s (alpha joint), 4.6 s (beta joint), and 4.2 s (gamma joint), see Table 2). The corresponding lower graphic, for comparison, shows the result of the situation where both—rhythmic sensory input as well as rhythmic input from the oscillation produced by the pilocarpine stimulation—are superimposed. To gain some variability, we used different periods for the sensory input. Fig 6A shows a CS period of 3.6 s and Fig 6B a period of 10.8 s (uppermost line). The output data are shown in colors as used in the earlier figures for Protractor–Retractor, Levator–Depressor, and Flexor–Extensor. Note, that due to the reset effect of the CS input, the exclusively pilocarpine driven cycles (above) and the combined data are in-phase only for the first period.

As mentioned, in Akay et al. [20] and in Hess and Büschges [35] rhythmic sensory stimulation has been chosen using shorter periods than those of the pilocarpine driven oscillations. In the simulations, we however also test cases where pilocarpine driven periods are shorter than sensory driven periods, which may then be used for further biological studies as a test. Due to the selection chosen, each of these two experiments (Fig 6A and 6B) contains altogether six different tests (2 CS cycles x 3 pilocarpine driven cycles). As a result, in all three joints of Fig 6A and the alpha joint of Fig 6B, the simulated data agree with the biological data as the CS input dominates all four cases. In Fig 6A, in all cases the CS period is shorter than that of the pilocarpine induced periods. In Fig 6B, alpha joint, the CS period is by about 6% longer than the pilocarpine driven period, but still the CS signal is dominating. In the beta joint and gamma joint

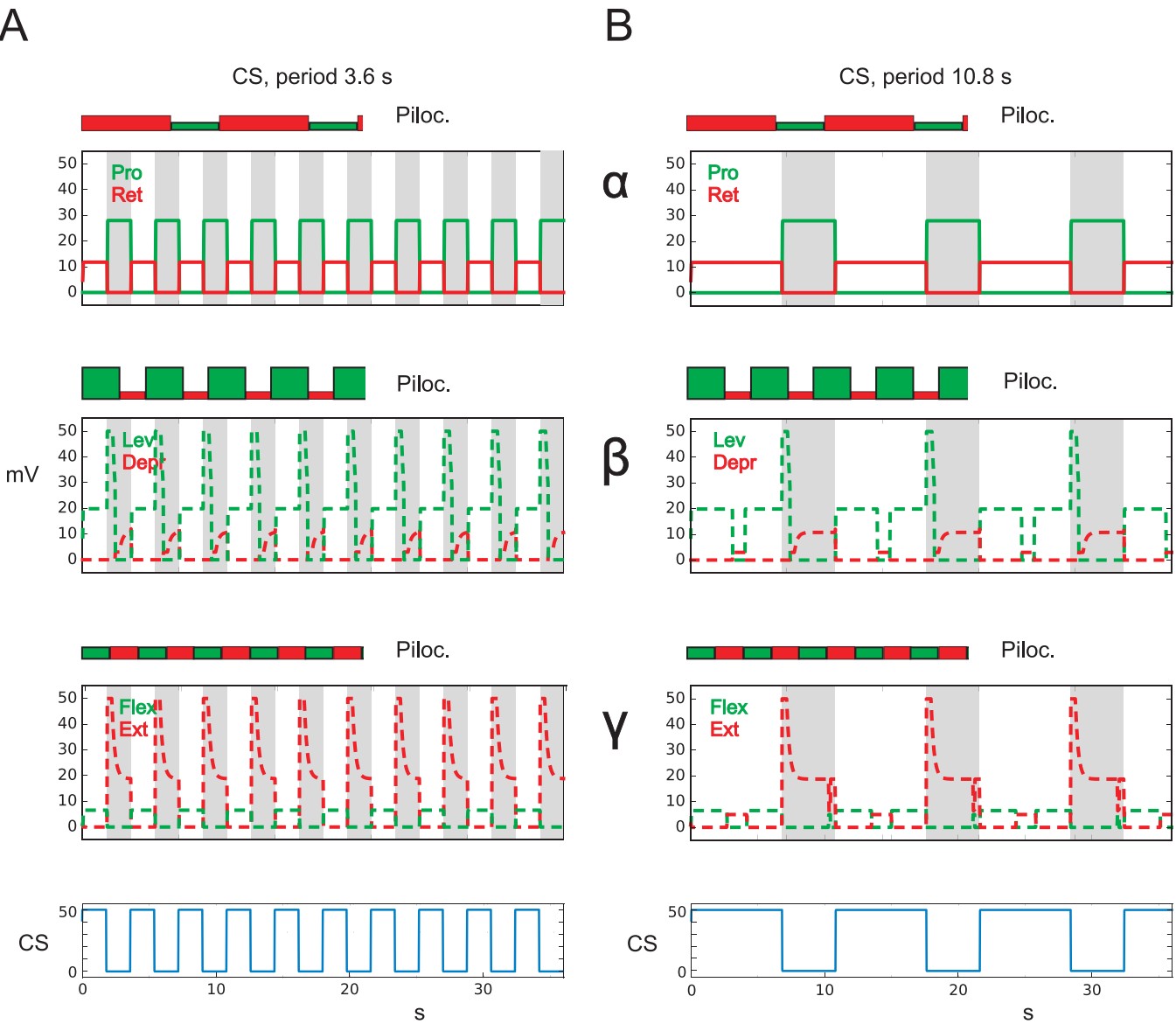

**Fig 6. CS stimulus drives (via caudal bending of femur) protractor or retractor as observed in Akay et al. [20].** Cycles that are driven only by pilocarpine are plotted for each joint on top of the CS-driven results (same scale, periods: alpha joint: 10.2 s, beta joint 4.6 s, gamma joint: 4.2 s) and marked by the same colors but with filled rectangles and 'Piloc.'. A) Sensory input, period: 3.6 s, B) Sensory input, period: 10.8 s. Abscissa: time (s), ordinate (mV). Dashed lines: simulation data for which no biological results are given.

of Fig 6B, the CS cycle only partly dominates, as the CS cycles are much slower than the corresponding pilocarpine driven cycles (sensory period: 10.8 s, for the beta joint: pilocarpine period 4.6 s, for the gamma: pilocarpine period 4.2 s). Therefore, the faster pilocarpine cycles dominate within the ongoing CS period, but still, in both cases, the changes of the CS signal nonetheless reset the output values.

Details biological experiment 2: As another, second biological experiment [35] treated the middle leg with pilocarpine ([35], their Fig 8), which elicited oscillations in the levator-depressor system, i.e., the beta joint [3]. Rhythmic or arhythmic fCO stimulation—introduced by elongation or shortening of fCO apodeme, corresponding to flexion or extension of the

gamma joint—was able to drive activation of levator muscle and depressor muscle, respectively. This was found within a limited frequency range.

Details simulation 2: Corresponding simulations, now following the experiments of [35], are shown in Fig 7 in the same format as used in Fig 6. In Fig 7A, the sensory input (bottom row, dark blue line) was applied using shorter periods (3.6 s) compared to those of pure

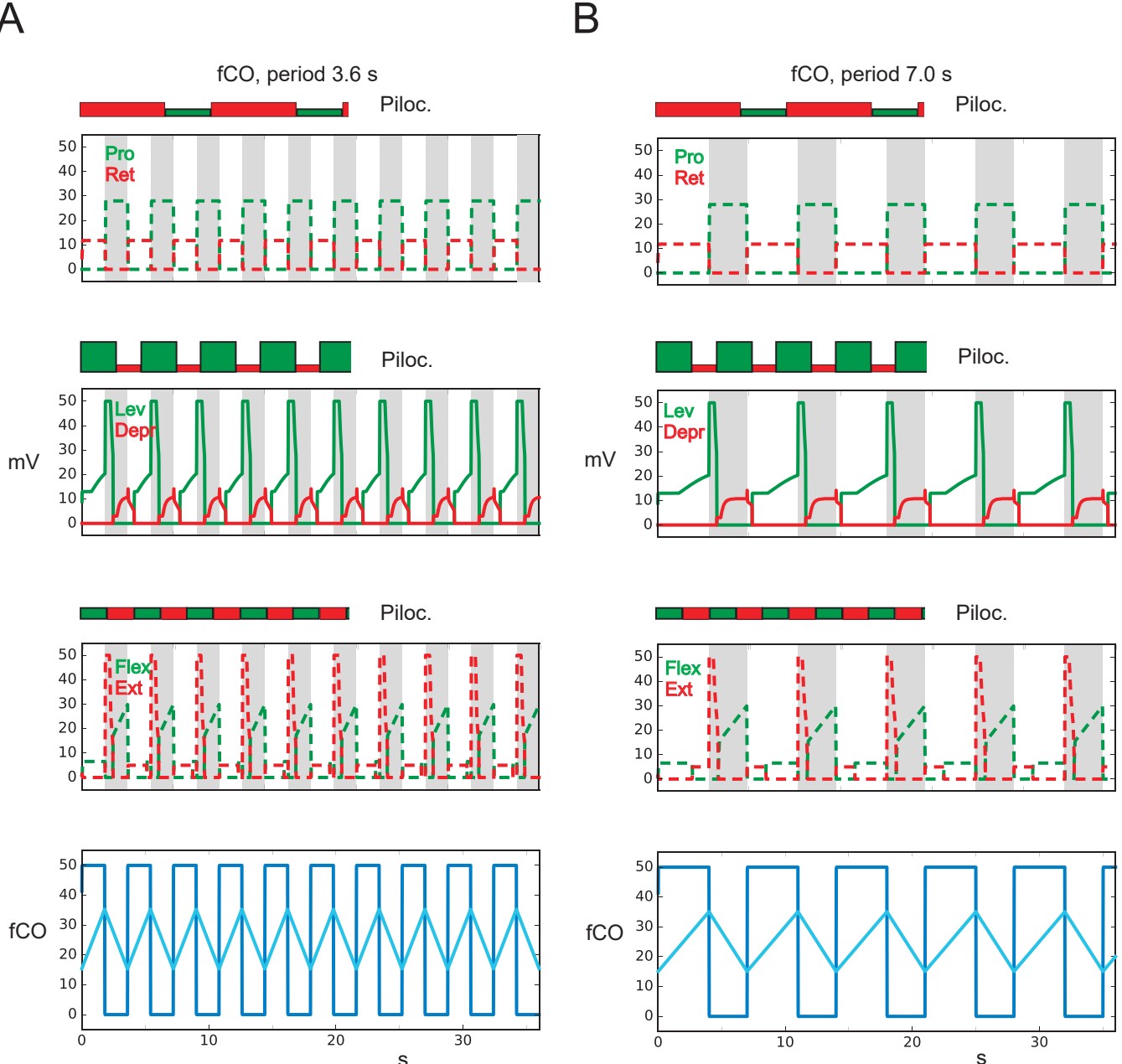

**Fig 7. fCO stimulus (elongation, shown in bottom row as light blue line) drives Levator or Depressor as observed in Hess and Büschges (1999).** Cycles driven by pilocarpine only are plotted for each joint on top of the fCO-driven results (same scale). The former are marked by the same colors but with filled rectangles. A) Sensory input, period: 3.6 s, B) Sensory input, period: 7.0 s. Abscissa: time (s), ordinate (mV). Dashed lines: simulation data for which no biological results are given. Gamma flexion corresponds to fCO apodeme elongation, fCOpos: position input to Levator-Depressor joint (gamma joint), light blue line indicates fCO apodeme position, fCO: (velocity) input to unit swing or unit stance (dark blue line).

pilocarpine-driven periods (data as in Fig 6A: 10.4 s, 4.6 s, 4.1 s, for alpha joint, beta joint and gamma joint, respectively). Therefore, as in the biological data, in all three joints pilocarpine-elicited oscillations are dominated by sensory influences. Fig 7B shows the corresponding experiment but now with longer sensory period (7.0 s, bottom row, dark blue line). The alpha joint is still fully controlled by the sensory input, as the sensory cycle remains shorter than the pilocarpine driven period (7.0 s vs.10.4 s). This is different for the beta joint (pilocarpine period: 4.6 s), and for the gamma joint (pilocarpine period: 4.1 s). In both these cases, influences from the pilocarpine-driven system can be seen. As in the biological experiments, reset effects can be observed in each joint. As a further effect, in both Fig 7A and 7B, in the gamma joint the first half cycle (the pair of Flexor/Extensor) shows an inverted temporal sequence compared to later cycles. Thus, as shown in the earlier experiment (Fig 6), sensory input may dominate slower pilocarpine-driven cycles, and resets the pilocarpine cycles if the latter is longer. To summarize: the simulation results show qualitatively the same results as observed in the biological experiments.

## 4 Discussion

In an earlier study [1], we developed a minimal architecture called neuroWalknet, a simulator that is able to emulate diverse behaviors of stick insects but might also in part be generalized to other insects. That study focused on behaviors concerning coordination of legs and interleg control as are forward and backward walking, a broad range of velocities, various—including "non-canonical"—gaits, negotiation of curves, climbing on irregular ground, but also partial or complete deafferentation of legs. We used an embodied simulation (i.e., with a dynamically simulated or physical body), and introduced decentralized structures that enable control of the individual legs. In the current study, we now ask to what extent a number of influential experimental studies which specifically deal with *intraleg* control, could be explained by the control structure neuroWalknet, too. The studies in this article have in common that sensory stimulation in a leg directly causes a local reaction in the different joints. The studies cover stimulation of CS affecting the alpha joint [19–21], stimulation of femoral fCO affecting the beta joint [35] or the gamma joint [17, 23–26, 31, 32]. As a further group of experiments, the influence of pilocarpine to control of movements of the alpha joint or the beta joint was studied. To represent these specific paradigms as states in the network, three pairs of motivation units (Fig 1A) have been introduced to neuroWalknet as an extension and are used to control different contexts on different levels.

To simulate all the experiments with neuroWalknet—as a general approach used in simulation studies—the structure is simplified compared to that of the biological system. In particular, the number of artificial neuronal units is much smaller than that of the biological counterpart, for there is only one instance of velocity unit (but see [37, 38]), one unit for backward walking and for forward walking each (but see [39–41]), and only two antagonistic motor units for each leg joint.

As a result, the behavior found in all groups of biological experiments could be qualitatively replicated and explained by the simulation. The simulation results in addition predict data for all three joints (only one joint has been studied in the biological experiments).

As one interesting behavior, we dealt with the so-called **Active Reaction**. Active reaction has been introduced as a reaction opposite to the well-known resistance reflex. A resistance reflex can be observed when an external mechanical influence, e.g., to the leg, is counteracted by the muscles being stretched. In contrast, in state active, the insect eventually shows an assistance reflex or active reaction: the muscle's reaction does not oppose, but support the

movement elicited by an external stimulus. However, as found in the biological results discussed above, such an active reaction does not occur in all cases.

As the effect can be observed in the open loop experiment, we conclude that the behavior called Active Reaction is not depending on body mechanics or embodiment, but depends on the neuronal structure of the leg controller (output velocity), and leg positions and walking direction. Further, the intensively studied Active Reaction [17, 23–25] has previously led to the interpretation that the Active Reaction requires an additional specific control structure acting as a positive feedback element used in specific situations only. As a consequence, this interpretation led to simulation concepts as developed by Schmitz et al. [42] or Goldsmith et al. [43]. However, the current results offer a simpler interpretation following Bässler's [17] postulate of a more general structure. In this alternative approach, neuroWalknet simply produces a velocity signal as assumed by different studies (e.g., [44–47]). This global feedforward velocity signal will then be controlled by states Walk, Forward or Backward, Stance or Swing and via walking direction "theta" by the ring net. This structure produces results that correspond to observations that can been called Active Reaction and includes an Active Reaction-like behavior also for the alpha joint ((Bässler, 1986), see Fig 3A). Furthermore, in some situations, no Active Reaction was observed in the biological experiment which agrees with our simulation results, too (e.g., Fig 3B). As an example, during negotiation of curves, the ring net in neuroWalknet provides an Active Reaction-like movement for the inner leg, but not for the outer leg. The letter appears as either a passive reflex-like behavior or a response that could be seen as a neutral reaction (between active and passive reflex). As another explanation, the effect simply follows the general structure of neuroWalknet depending on the current context. Our results (Fig 4) correspond well to the biological experiments.

In other words, in this view Active Reaction does not represent a dedicated neuronal structure, but a phenomenon that eventually emerges in specific contexts.

The details as to how the information given by the sensory input might be transmitted within the neuronal system is still open. In a recent article, Goldsmith et al. [43] proposed the idea, supported by their quantitative simulations of a leg model, that the desDUM neurons [48] may provide a significant contribution and they further argue, based on studies of Sauer et al. [49] and Driesang and Büschges [50], that a switch between states Walk and Stand on the local level may at least partly depend on changes of non-spiking interneurons.

Generally, these new results provide another example demonstrating that complex behavior does not necessarily require complex neuronal structures (see also Navinet [15, 16], positive feedback-based walking [42, 51]).

Open questions: An open point concerns the implementation of the neural structures that represents the effect of caudal bending [20, 21], or the effect of ramp-like stimulation by elongation of the femoral fCO receptor apodeme [25, 26], that may lead to activation of unit Stance (or unit Swing). For caudal bending, the corresponding sensors seem to be represented by several groups of trochanteral CS [30] and it is even less known how exactly the ramp signal of the fCO is transformed into an appropriate signal that activates state Stance. Correspondingly, the neuronal structures that connect stimulation of abdomen or antennae with unit FW or BW are unknown.

Another difference between legs has to be mentioned. In the experiments of Akay et al. [20] CS stimulation by bending the femur elicited retractor activation in forward walking and protractor activation in backward walking. This is different in the hind legs, but only when all other legs have been deafferented. This observation would support the idea [32] that the isolated hind leg has an inherent tendency to walk backwards, a concept not yet implemented in neuroWalknet.

Central Pattern Generators: As shown in the results, Figs 6 and 7, we simulated neuronal systems [20, 35] that were stimulated not only by sensory input (CS, fCO), but were in addition treated with pilocarpine, which leads to oscillations in the antagonistic structure of each joint. This is interesting because it is often assumed that such—postulated—CPGs form the basis to control quasi-rhythmic walking whereby these rhythmic elements are modulated by sensory input (e.g. Akay et al., p. 3285 and refs.), i.e., CPG activations are assumed to operate during walking even if no pilocarpine is applied. Our results using a simpler control structure open the route for a complementary role and an evolutionary path. Early-on, inhibitory connections evolved in order to prevent co-contraction of antagonistic muscles. Remarkably, already these simple structures as found in neuroWalknet show oscillatory properties that improve locomotion behavior and lead to stable cyclic movements. These capabilities were demonstrated in our original simulations [1] and further in current experiments (see Figs 2 to 4). The current simulations (Figs 6 and 7) demonstrate that the pilocarpine-driven experiments allow for oscillating motor output in neuroWalknet as well, again without an explicit CPG structure and that the structures in neuroWalknet that show oscillations driven by the pilocarpine input can be dominated by sensory input, or can reset, as shown in the biological experiments. Only later-on the advantageous oscillatory properties evolved into explicit structures controlling oscillations that introduced additional connections.

Usually, these two types are treated in the literature as alternatives. On the one hand, in a CPG based account, it is assumed that in each joint there exists an internal, explicit CPG structure that is used for walking and may also be activated when treated with pilocarpine. Such a structure may correspond to a combination of both structures shown in Fig 5, with (a) inhibitory neurons equipped with phasic properties (as in Fig 5, left, PMN), but in addition (b) a common input that, for example, may be connected with unit Walk, and even further (c) be connected with pilocarpine inputs. On the other hand, in neuroWalknet as a second hypothetical network, there is no such internal CPG structure, but only the structure as shown in Fig 5 (left, PMN), that (a) contains inhibitory neurons with phasic properties and (b), through external input from pilocarpine, may produce oscillations when activated. In this case, the internal structure as such does not allow for oscillations, but application of (simulated) pilocarpine (Fig 5, left, purple arrows) may trigger the network to start oscillations. Thus, in this case the experiments shown in Figs 6 and 7 could be explained without a dedicated CPG structure being necessary. This is important, as often CPG-like structures are assumed as minimal and required. The simpler neuroWalknet structure provides an alternative explanation as evolved for slow walking.

While the morphological differences between both these hypothetical networks are small, there is an overall functional difference. neuroWalknet provides a functional advantage as no taming of quite a number of CPGs is required when stopping or when dealing with irregular substrate and negotiation of tight curves.

Importantly, our assumptions on neuroWalknet are not meant as a claim that there may no other CPGs being active in stick insects. The study of rocking behavior [52], for example, demonstrates the existence of CPGs being present. Another suggestion comes from Bässler et al. [53], who studied searching movements (for details see Supplement). They observed leg oscillations of up to 10 Hz, as well as similar data from Drosophila [54], which also suggests the involvement of dedicated CPGs. Interestingly, Stoltz et al. [48] studied influences of CS on stance activation of walking leg via desDUM neurons situated in the gnathal ganglion, but did not find influences of pilocarpine induced CPGs, which may support the idea that pilocarpine induced CPGs are not involved for control of slow walking. Similarly, Mentel et al. [55] found DUM cells in the mesothoaric ganglion most of which were active during stance phase of the

leg studied, but only one further unit (DUMna nl2) exhibited spontaneous rhythmic activity. However, this rhythm was not coupled to leg activity.

## 5 Conclusion

Taken together, neuroWalknet is considered a holistic approach that combines important behavioral elements within one complete system with the hope to better recognize emergent properties of the system. We believe that such an approach leads to better understanding of the complete system, rather than considering only separate, local elements.

We further believe that neuroWalknet represents an RNN structure that combines the essential properties required for legged locomotion in insects (or, maybe, even for arthropods). The basic structure consists of an RNN called motivation unit network (Stand-Walk, FW-BW, Stance-Swing) controlling sub-behaviors as are: Walk: curve, and swing or stance; Swing: swing movement and search; Stance: stance movement including height control, i.e., obstacle avoidance, negotiation of curves; Stand: Operation with negative position feedback including adaptive setpoint depending on substrate compliance (addressed in the Supplement).

Whereas some of these behaviors have been simulated, but not yet been integrated into neuroWalknet, the most relevant ones are represented by network structures that (i) allow for quantitative simulation of the behaviors, but may (ii) also serve as a scaffold to test new hypotheses concerning not yet considered behavior of alive animals, also with respect to other species. And (iii) allow the (simulated) robot to predict behaviors that have not yet been tested biologically (as, e.g., shown by data in dashed lines, Figs 2–4, 6, 7)).

Important, but still open questions refer to (i) a possible neuronal realization of the ring structure (see [56]), (ii) control of segregated hindlegs, (iii) proof of the existence of CPGs to be implemented during walking, and (iv) how to establish an appropriate solution for simulation of searching movements.

## 6 Methods

We developed a simulation model in order to better understand the function of a complex system and to provide predictions that could be tested in later biological experiments [57]. Simulation models purposefully focus on specific, selected aspects and abstract away other details. A given model always introduces simplifications and every model is in one or the other sense "always wrong" [58] as they focus on essential properties only [59]. For instance, in our simulation the number of artificial neurons, for example motoneurons, is much smaller than that of the corresponding biological system. Likewise, the number of load sensors and their functional details are extremely simplified (Fig 1A, turquoise units) compared to those studied by [30]. Overall, models are a useful tool in computational neuroscience as they aim to summarize functional relations and aspects, provide explanations, and lead to falsifiable predictions [59].

The control model has been introduced in detail in [1] and we have explained extensions in section 2 and changes in this section. As a brief summary: The simulations consists of two main parts: On the one hand, the neuronal controller processing sensory inputs and producing control signals on a per leg basis (the neuroWalknet controller has been implemented in python (version 3), see https://github.com/hcruse/neuro_walknet and there the 2022 version). On the other hand, a dynamic simulation environment for the body of the hexapod robot Hector [2, 60], which exists as a hardware version and as a dynamic simulation (implementations are publicly available: dynamical simulation environment is realized in C++ and based on the Open Dynamics Engine library, see https://github.com/malteschilling/hector). Here, we use the dynamic simulation.

The neural controller consists of artificial neurons. While the activation dynamics of neurons are often approximated using Hodgkin-Huxley differential equations [61], a reduced version has been used for simplification [62].

The structure of neuroWalknet is decentralized: There is one identical control network for each of the six legs (see Fig 1A). Each of these leg controllers consists of three subnetworks (Fig 1A, bottom three rows, marked as alpha, gamma, beta), one for each joint. In addition, there are a number of global units that provide context information and allow to switch on or off walking and for selecting walking direction. Furthermore, there are units for interleg coordination realizing the coordination influence rules. On the motor level, the output is simplified as we consider for each joint two antagonistic muscles. For more details on neuroWalknet, see [1].

Another relevant aspect concerns the structural basis of so-called central pattern generators, being studied in a specific set of biological experiments where the neuronal network is treated with pilocarpine. In such experiments, usually sensory input and motor output channels and specific connectives are cut [3]. When treated with pilocarpine, the motor outputs of each of the three leg joints oscillate rhythmically, usually with a period in the range of 4 s to 10 s. These cycle frequencies are quite variable (they depend on pilocarpine concentration and probably on further parameters, as well as on the leg joint, see Table 2). As different studies show quite different values, in neuroWalknet we based our experiments on data given by [3] (Table 2, 1st line, 'BSB'), as they provide the most solid data base and used similar periods in our simulation (Table 2, line 4, output pilo; for comparison: walking periods of *Carausius morosus* operate in the range of 0.5 s and 3 s [63]).

In the biological experiments used for comparison in experiment four, specific sensory inputs have been left intact. As a consequence, interesting interactions can be observed when sensory cycles take about equal the time of the induced oscillations or a bit faster than the cycles elicited by pilocarpine only (e.g. [20, 35]). To study this effect in simulation, we decided to apply artificial pilocarpine signals (Table 2, 3rd line) with a frequency range corresponding to that of slow robot walking: Pilocarpine-driven cycles have been tuned to a period between 2 s and 10 s in the simulations (Table 2, 4th line), i.e. these cycles are in the same range as we have used in earlier experiments [1], see Table 2, 7th line.

We selected input values in simulation that allowed some variability and cover a sensible range of periods. This was realized by selecting different sensory input patterns: these cyclic sensory stimuli (CS in Fig 6, fCO in Fig 7) consist of two parts, with a longer and a following shorter time window (the latter marked grey. Whereas in the earlier experiments not using pilocarpine (Figs 2–4) these time slots concern 4 s and 2 s respectively, i.e. a period of 6 s, in the experiments where pilocarpine is given (Figs 6 and 7), different time slots for sensory stimulation have been tested and two interesting examples are shown (Fig 6, period 3.6 s and 10.8 s; Fig 7, period 3.6 s and 7.0 s.

Table 2 compares duration of the six motor output values, i.e. the period, for data elicited by pilocarpine only ([3], Table 2, 1st line, 'BSB), and used for the current simulation ((Table 2, 4th line 'output pilo'). The corresponding periods are of about the same order in all three cases. However, the input values, i.e., the strength of the pilocarpine signal that elicits these periods (Table 2, 3rd line), are smaller for the current version than those used in [1] (Table 2, 6th line). This was chosen because higher input values could have dominated the effect of the sensory input.

## Supporting information

**S1 File. Additional Experimental Data for Standing and Searching Movements.**
(PDF)

## Author Contributions

**Conceptualization:** Malte Schilling, Holk Cruse.

**Data curation:** Malte Schilling.

**Formal analysis:** Holk Cruse.

**Funding acquisition:** Holk Cruse.

**Investigation:** Malte Schilling.

**Methodology:** Malte Schilling, Holk Cruse.

**Software:** Malte Schilling.

**Writing – original draft:** Malte Schilling, Holk Cruse.

**Writing – review & editing:** Malte Schilling, Holk Cruse.

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
