## [Decision Letter · Decision Letter 0]

15 Jul 2022

Dear Dr. Schilling,

Thank you very much for submitting your manuscript "neuroWalknet, a controller for hexapod walking allowing for context dependent behavior" for consideration at PLOS Computational Biology.

As with all papers reviewed by the journal, your manuscript was reviewed by members of the editorial board and by several independent reviewers. In light of the reviews (below this email), we would like to invite the resubmission of a significantly-revised version that takes into account the reviewers' comments.

As you will see, the reviewers are generally enthusiastic about your work, but also raise several important issues that need to be addressed before a potential publication.

A common theme between reviewers is a lack of clarity in the manuscript text and the some of the figures, making the manuscript logic difficult to follow. This corroborates my own assessment of the manuscript. The reviewers provide a list of suggestions that may help address this issue, but i would urge you to critically revisit some of the text to clarify the approaches and the logic behind them, and to make the manuscript accessible to audiences outside of stick insect locomotor control.

A second major point raised by all reviewers is the potential presence or absence of CPG or CPG-like architecture and activity, and its impact on your results and conclusions. CPG vs feedback has been an ongoing discussion for decades now. CPG-like rhythmic activity has been identified in experiments with animals, and your model contains reciprocally inhibitory circuits that may resemble CPGs. It seems important to be explicit about the conditions under which you observe CPG-like activity in your model, and what you conclude from these experiments. The reviewers provide several starting points for a discussion of CPG vs. non-CPG structures and their influence of the resulting behaviors.

We cannot make any decision about publication until we have seen the revised manuscript and your response to the reviewers' comments. Your revised manuscript is also likely to be sent to reviewers for further evaluation.

Sincerely,

Wolfgang Stein

Guest Editor

PLOS Computational Biology

Samuel Gershman

Deputy Editor

PLOS Computational Biology

Reviewer's Responses to Questions

**Comments to the Authors:**

Reviewer #1: This is interesting work but the paper needs substantial revision for clarity.

The core aim of the study is to show that a decentralized neural network for hexapod walking, which has previously been shown to effectively reproduce a number of characteristics of walking behavior in insects (specifically stick insects) can also be shown to reproduce some widely studied phenomena observed in single leg reflexes. This is significant because these phenomena have in the past been taken to be indicative of particular neural control mechanisms but might in fact be just emergent properties under rather specific experimental conditions. The work presented also allows predictions of some new effects that could be observed. Finally, testing the model under these conditions, which it was not explicitly developed to explain, should provide good evidence that it may indeed be a correct account of the mechanism in the insect.

However, the final point to some extent depends on how clear it is that the ‘minor (technical) expansion’ (line 127) to the previous model that has been made in this paper is not in fact careful tweaking to the model designed to produce the desired outcomes under these new conditions. E.g., the fact that the model required some additions to allow the new experiments is not even mentioned in the abstract. Also, several of the experiments being reproduced seem quite complex in terms of the controls applied to the animal (isolated legs, removed sensing, no ground contact, etc.) and it is frequently hard to follow exactly how those conditions have been applied to the model, or how much scope there is within the model to vary the exact implementation until the desired qualitative effect emerges. I think the authors can probably defend their approach against such criticism, but the paper needs to be better presented to do so.

A final general comment is that I felt the writing could be significantly improved. In particular there seems to be a lot of repetition, with the same point being made several times in subsequent paragraphs. This is often confusing as the reader is not sure if it is the same point or a new (related) point that needs to be understood before proceeding. I have endeavoured to note some specific examples of this problem below. The text also seems to fall between the gap of not explaining enough about the model to be comprehensible without reference to the previous paper, yet explaining too much to be read as a straightforward extension of the work in the previous paper, as it seems to repeat a number of points already made in that paper. Most specifically, the demonstration that ‘CPG-like’ oscillation can occur in the decentralized network under conditions taken to mimic the application of pilocarpine is shown inFig 8 in Schilling & Cruse, 2020, but the current paper is not always clear as to whether the observation of emergent oscillations is a new result here or not.

Specific comments

Line 22, “Furthermore, such reflexes have been studied while the ganglion was treated with pilocarpine” - as far as I could tell from this paper, the pilocarpine experiments differ from the reflex experiments, they are not a variation of them.

Line 29 “When experimenting with pilocarpine” - this is an example where the paper sets up the expectation in the reader that the appearance of oscillations is a novel result; and does not refer to what I take to be actually new in this paper, i.e., showing entrainment of these oscillations to sensory stimuli.

Line 67, it seems odd that the ‘answer’ to the ‘first question’ “how could possible functional elements [of a control system] be defined” is to say, in effect, ‘we will just use some elements we defined earlier’. I find the whole way this part is introduced somewhat awkward, and would prefer they went straight to the point (line 71) that the purpose of this article is to test an existing model against some different behavioural findings.

Line 94, line 98, line 108 example of unnecessary repetition - “Specifically, the focus will be on context dependency” “will in particular focus on context dependency” “In all these experiments the response depends on the context”

Line 97, I think it would be helpful to the reader to explain clearly each of the reflexes to be studied at this point in the paper, rather than describe one and hint at the others. In particular it seems important for motivating the work to explain that the ‘active reaction’ effect has been considered to require explicit mechanistic control, and hence it is very relevant to understand if it might just emerge from the current model. If possible, some kind of illustrations to accompany table 1 would help a reader less familiar with the system to grasp the key points for each experiment. Also explaining the range experiments here, and then, in the relevant results section only briefly recapitulating the specific experiment being reproduced in simulation would make it much easier to follow the results.

Line 111 “behaviour of a (more or less) intact animal has been compared with animals experiencing the same treatment but, in addition...” - what treatment are these (more or less) intact animals experiencing?

Line 132 “Furthermore...” these two sentences seem unnecessary, the point has already been made on lines 120-122. Or remove lines 120-122.

Line 145 “While most control approaches employ one holistic control unit that controls all available degrees of freedom”. I am struggling to think of any control approach in either robotics or biological modelling that fits this description. It seems unnecessary to make this statement.

Line 152 on, it would help if this paragraph was written much more clearly as an explanation of what is common to Walknet and the current model, and what is different. E.g. is the line “Different actions are represented by so called Motivation Units” true of Walknet? As presumably it is not true in Walknet that these are “simple artifical neurons etc.”. I do not understand the sentence “In neuroWalknet, as the main difference, the control circuits as such have been included into these neural circuits, one for each joint.” Do you mean the controllers “have been implemented as neural circuits”; and is the property of “one for each joint” specific to neuroWalknet or inherited from Walknet?

Line 169 on, I ended up thoroughly confused about what was new to the model in this paper. Is it including pairs of states (stand/walk, stance/swing, forward/backward) that were in Walknet (line 174) but were not in the previous neuroWalknet (line 180)?

Line 196 on, is the point being made here that the current simulations are just going to look at the activation of the motor units, and not the actual leg motions produced? I actually found it quite hard to grasp from the paper whether the simulations had been done with full coupling to the simulated body of the stick insect (it appears this was the case from line 817 in the methods?) or how important this was to the results, given these are reported in terms of motor neuron activity only.

Line 219 on briefly describes the new specific sensory inputs introduced into the model, referring to the methods for details. However, it is not well explained here or in the methods (lines 833 on) how it was decided that the new CS inputs should connect to the Swing/Stance units or the fCO be connected to Stance or to the specific beta joint. Biological data may be lacking but is there a functional justification, beyond that of the specific results that this model is meant to explain?

Results

General comments: I assume the grey/white background to the figures represents swing/stance but this was not stated anywhere I could notice. It is potentially confusing to the reader why the various motor outputs start in such different states before the stimulus is applied. It would also help, for these figures, to include alpha/beta/gamma labels, as in the circuit diagram, for clarity. The captions unnecessarily describe details (what the line colours, and graphs in different positions show) which are labelled within the figures; while there is not enough in the captions to highlight the key observations made. Throughout, I felt it might have helped to have an accompanying illustration of the experimental procedure that is being replicated, e.g. showing the animal is intact or missing legs or sensory feedback, where the stimulus is applied, etc. I did not understand why the results for passive state, which are frequently presented first in the sections describing the biological phenomena were not included in the main text of the paper.

Fig 2, text from 281, it is not fully clear how the state is set up in the model to be a close reflection of the experiments, but it seems the result is a straightforward consequence of having set up the CS input to trigger transition to stance, plus the opposite direction of motion in stance in backward walking. The most conspicuous change is the decrease in Pro-output (Ret) in forward (backward) walking but this is not mentioned in the text. Line 288 “in both cases, relaxation of CS stimulation leads to activation of the antagonistic muscle” - is this in the simulation or the animal or both?

Fig 3 and related text, why not use the middle leg in the simulation, given that this seems to be the crucial leg in the experiments? It is not very helpful here to label the blue line as FeTi flexion when everywhere in the text this is referred to as the gamma joint.

Fig 5 appears to be the same data as fig 4, only to demonstrate matching of a different biological observation. Is there any reason to keep these separate, rather than just to show in fig 4 that there is a match to existing data on the beta joint as well?

In the interests of a timely return to this review, I will just note that the discussion again contains substantial repetition and could be much more concise.

One final specific conceptual comment: I am not sure it is necessary to maintain such a strict dichotomy regarding existence/non-existence of a CPG. Indeed, this study and the previous paper support the idea an evolutionary process: first the animal develops a mutually inhibitory network to prevent opponent muscle co-contraction; then it has the possibility to co-opt the implicit oscillatory properties of this network to improve control.

Reviewer #2: This excellent and ground breaking paper examines the potential mechanisms underlying pattern generation and motor control in stick insect walking, focusing upon the context dependence of sensory feedback. The studies use the artificial nervous system, neuroWalknet, including modifications such as local pre-motor interneurons with reciprocal inhibitory connections (in premotor and 'motivational' neuron nets). In a tour-de-force series of tests, the model is used to examine the results of a number of previous biological experiments. Those studies include: the classic reflex reversal and joint specific effects elicited by stimulation of the femoral chordotonal organ; context dependence of sensory feedback in curved walking; and, remarkably, the effects of sensory feedback from sensory receptors indicating force and those signaling kinematic variables on rhythmic activities artificially induced by pilocarpine. In each case, the model is able to reproduce the major findings of the studies and also provide insight into the underlying mechanisms that could be utilized and tested in future biological experiments. Although the details of the reasoning behind each simulation can be discussed and 'picked apart' by others in review, I believe the model results, as whole, are remarkably accurate in reproducing the biological data and generate new testable hypotheses.

This is one of the most important papers to appear in the field of pattern generation and motor control in recent years and offers a different perspective than other studies. While the authors extensively discuss the issues of the existence of CPGs, which is conceptual and open to diverse interpretations, I strongly believe that the paper should be published as it offers new approaches that can be tested experimentally. This is a major advance in the field.

I have suggested a number of revisions and corrections to the text that I feel will strengthen the manuscript but use of the suggested text (below) is at the discretion of the authors and editor.

Major (significant) revisions

1. Changing the emphasis of discussion of CPG's - The descriptions of the simulations reported in this study are excellent. However, the discussion of these results focuses upon how the proposed model does not contain elements that are intrinsically rhythmic, i.e. the stimulation does not rely upon a CPG. While that reasoning is valid, I believe that the issue of whether interneurons in the CNS can be conceptually described as a CPG is secondary to the major strength of the paper, i.e. formulating an artificial nervous system that can accurately reproduce a broad range of biological data. I suggest that including elements of the following in the discussion:

'These results show that formulation of a CPG is not required to reproduce the results of a number of neurobiology experiments done in stick insects. However, our simulation does contain postulated elements and connections that are often included in simulations of CPGs, i.e. mutually inhibitory interactions between premotor interneurons that can activate groups of antagonist muscles. In our formulation, these elements do not exhibit rhythmic activity in isolation although they are apparently essential for generating the 'rhythmicity' that occurs in walking. The major difference between our findings and other formulations is the word 'central' in CPG. CPGs are often depicted as discrete elements contained in 'black' boxes in the nervous system that can function in isolation (or with limited sensory inputs). In contrast, we have 'filled the black box' and described model pre-motor interneurons with discrete properties. Further experiments are necessary to study and characterize spiking as well as non-spiking interneurons in behaving (walking) animals to test whether similar interneurons can be identified in the stick insect CNS.

2- Critical analysis and modifying discussion of caudal bending - line 208 and elsewhere - It is important that the authors include the following caveat in discussion of experiments using caudal bending of the femur. Although it has been used experimentally in a number of biological experiments, the natural circumstances in which caudal bending would occur are unclear. CS respond to resisted contractions of muscles and caudal bending (simultaneously activating trochanteral and femoral groups of CS) would only result from a very large resisted contraction of the protractor muscle (i.e., encountering an obstacle during the swing phase.) It has been argued that caudal bending occurs transiently at the onset of stance after the tarsus makes contact with the substrate. The only evidence to support that hypothesis is ground reaction forces that are much smaller and shorter in duration than the forces applied experimentally. In all, the findings of massive and potent effects of caudal bending applied to the femur are likely artifactual and do not occur in nature. Although the present study does not directly address these issues, it is important that the authors critically discuss the experiments used in simulations.

Minor corrections

Abstract - line 14 - better as: This controller allows for walking patterns at different velocities in both forward and backward directions.

line 20 - better as: The reflexes are elicited by stimulation of the femoral chordotonal organ (fCO) or groups of campaniform sensilla (CS).

line 29 - The current statement is in a sense misleading and I believe that the authors intend: In simulations of the effects of pilocarpine.

line 75 - better as: Indeed, it allows for control of the basic different aspects of forward and backward walking,

line 88 - clearer as: this would further support the validity of the model.

line 144 - delete words: the case of

line 149 - delete words: still it is shown that

line 155 - Change 'a leg' to 'that leg'.

line 631 - better as - we developed neuroWalknet, a simulator that is able emulate diverse behaviors

Reviewer #3: The authors actual work represents the further evolution of the Walknet architecture, born on the basis of observations and experiments on the stick insect and now a paradigm for the decentralised control of locomotion, apparently in contrast with the centralised locomotion control, namely CPG. The set of basic rules governing interleg motions, strictly depending on local feedback, later evolved in neuroWalknet in 2020, which took into account other behavioral and physiological results, while negotiating curves or changing locomotion pattern and velocity, stopping and so on.

In this manuscript the attention is focalised on interjoint (intraleg) relations, relying on specific experiments based on stimulation of specific interjoint sensors, like femoral Chordotonal organ or campaniform sensilla. It is shown that the simulated neuroWalknet can reproduce the corresponding experimental results, and generalise them to other legs or joint motions, not yet detected experimentally, working as a simulation/prediction setup to be tested in future experiments on insects.

Moreover, neuroWalknet, at the level of intraleg joint evolutions, can show also context-dependent behaviors.

The paper is nicely written and experiments, as well as simulations, are sometimes so detailed that it is somewhat difficult to follow, at a first reading.

Some main points need to be fixed in order to further refine the paper:

The first one regards the concept of active reaction vs. resistance reflex. As outlined by the authors, resistance reflex, (i.e. negative feedback) or active reaction depend on the insect state (active or passive), and are context-dependent. The authors should specify if the active reaction may also depend on the coupling between neuroWalknet and the dynamically simulated body. This is unclear, since the authors declare that all the simulations were done in an open loop. Is this related only to the stimulation of the specific sensors (femoral Chordotonal organ or campaniform sensilla) or all the simulations are performed without a “body”?

The second point is that, since the network was built from a decentralised, feedback-dependent point of view, the authors conclude that there is no need for a central controller for locomotion, as instead widely accepted. Indeed neuroWalknet structure contains basic modules, the “bistable monopoles” organised to control the motor neuron activity, with the ability to generate oscillations in presence of particular stimulations.

If from the one side, CPG, that was born essentially as an open loop control scheme (see Pearson’s or Grillner’s works) a form of sensory feedback was needed to include to account for necessary adjustments coming from the environment.

On the other side, the concept of pure decentralization should be relaxed towards a more flexible architecture, allowing for the possibility to host endogenous oscillations (revealing the emergence of CPGs) which drive locomotion, for example when escape reactions have to be implemented. The neuroWalknet indeed contains such basic modules and the authors should further outline that this scheme would complete, (instead of compete with) the CPG scheme, which, under this view, would represent a specific solution of a wider locomotion scheme whose major assumption is the need for embodiment. Even if there is no learning within this structure, the CPG units would represent learned (fast) solution, to be implemented in specific conditions, within a wider architecture, like neuroWalknet. This issue could be discussed by the authors in the discussion Section.

Minor issues:

Fig.1: this figure represents the section devoted to the control of the joints of a single leg. However in the lower central-right hand side of the figure, the green nodes and the boxes are reported and nont mentioned. Please try to at least sketch the role and refer to the literature. S

ince attention is posed to the control of intraleg joints, a scheme of the leg, reporting the joint and the associated motion, as several times reported by the same authors in other papers, should be inserted into Fig.1. This would allow readers to follow the paper more fluently.

Lines 220 etc. please specify why the input “load” seems to have a stepwise form, whereas the fCO is graded, ramp-like. Is this an assumption for the simulation or it is suggested from experiments?

Line 282 “as soon as the leg is bend” � bent

The effect of pilocarpine on the PMN module would be better explained, possibly with a figure. In line 229 the authors report that the structure is a soft-WTA, but that a simultaneous input would onset oscillations.

**Have the authors made all data and (if applicable) computational code underlying the findings in their manuscript fully available?**

Reviewer #1: Yes

Reviewer #2: Yes

Reviewer #3: Yes

PLOS authors have the option to publish the peer review history of their article (what does this mean?). If published, this will include your full peer review and any attached files.

Reviewer #1: No

Reviewer #2: No

Reviewer #3: **Yes: **Paolo Arena
---

## [Decision Letter · Decision Letter 1]

18 Nov 2022

Dear Dr. Schilling,

We are pleased to inform you that your manuscript 'neuroWalknet, a controller for hexapod walking allowing for context dependent behavior' has been provisionally accepted for publication in PLOS Computational Biology.

Best regards,

Wolfgang Stein

Guest Editor

PLOS Computational Biology

Samuel Gershman

Section Editor

PLOS Computational Biology

The reviewers agree that all remaining questions and comments have been resolved.

Reviewer's Responses to Questions

**Comments to the Authors:**

Reviewer #1: The authors have done a thorough review which has answered my previous comments.

Reviewer #2: The Authors have addressed all the issues raised about the paper. This is a superb paper and it should be published immediately.

Reviewer #3: The paper has been revised as requested. The reviewer is satisfied.

**Have the authors made all data and (if applicable) computational code underlying the findings in their manuscript fully available?**

Reviewer #1: Yes

Reviewer #2: Yes

Reviewer #3: Yes

PLOS authors have the option to publish the peer review history of their article (what does this mean?). If published, this will include your full peer review and any attached files.

Reviewer #1: No

Reviewer #2: **Yes: **Sasha Zill

Reviewer #3: **Yes: **Paolo Arena

---

## [Editor Report · Acceptance letter]

11 Jan 2023

PCOMPBIOL-D-22-00647R1 

neuroWalknet, a controller for hexapod walking allowing for context dependent behavior

Dear Dr Schilling,

I am pleased to inform you that your manuscript has been formally accepted for publication in PLOS Computational Biology. Your manuscript is now with our production department and you will be notified of the publication date in due course.

With kind regards,

Zsofia Freund
